# COLLABORATIVE COMPRESSORS IN DISTRIBUTED MEAN ESTIMATION WITH LIMITED COMMUNICATION BUDGET

## ABSTRACT

Distributed high dimensional mean estimation is a common aggregation routine used often in distributed optimization methods (e.g. federated learning). Most of these applications call for a communication-constrained setting where vectors, whose mean is to be estimated, have to be compressed before sharing. One could independently encode and decode these to achieve compression, but that overlooks the fact that these vectors are often similar to each other. To exploit these similarities, recently Suresh et al., 2022, Jhunjhunwala et al., 2021, Jiang et al, 2023, proposed multiple *correlation-aware compression schemes*. However, in most cases, the correlations have to be known for these schemes to work. Moreover, a theoretical analysis of graceful degradation of these correlation-aware compression schemes with increasing *dissimilarity* is limited to only the $\ell_2$-error in the literature. In this paper, we propose four different collaborative compression schemes that agnostically exploit the similarities among vectors in a distributed setting. Our schemes are all simple to implement and computationally efficient, while resulting in big savings in communication. We do a rigorous theoretical analysis of our proposed schemes to show how the $\ell_2$, $\ell_\infty$ and cosine estimation error varies with the degree of similarity among vectors. In the process, we come up with appropriate dissimilarity-measures for these applications as well.

## 1 INTRODUCTION

We study the problem of estimating the empirical mean, or average, of a set of high-dimensional vectors in a communication constrained setup. We assume a distributed problem setting, where $m$ clients, each with a vector $g_i \in \mathbb{R}^d$, are connected to a single server (see, Fig. 1a). Our goal is to estimate their mean $g$ on the server, where

$$g \triangleq \frac{1}{m} \sum_{i \in [m]} g_i. \tag{1}$$

We use $[m]$ to denote the set $\{1, 2, \dots, m\}$. The clients can communicate with the server via a communication channel which allows limited communication. The server does not have access to data but has relatively more computational power than individual clients.

This problem, referred to as *distributed mean estimation* (DME), is an important subroutine in several distributed learning applications. Two common scenarios for these applications are distributed training, when different clients correspond to different processors inside a datacenter or federated learning McMahan et al. (2016); McMahan & Ramage (2017), when different clients correspond to different edge devices, for instance mobile phones. In distributed training, the communication channel is the network inside the datacenter, while in federated learning, the communication channel can be the internet.

The typical learning task for DME is supervised learning via gradient-based methods Bottou & Bousquet (2007); Robbins & Monro (1951). The vectors $g_i$ then correspond to the gradient updates for each client $i$ computed on its local training data and $g$ is the average gradient over all clients. On the other hand, distributed mean estimation is also used in unsupervised learning problems such as distributed KMeans Liang et al. (2013) and distributed PCA Liang et al. (2014) or distributed power iteration Li et al. (2021). In distributed KMeans and distributed power iteration, $g_i$ corresponds to estimates of cluster center and the top eigenvector respectively, on the $i^{th}$ client.

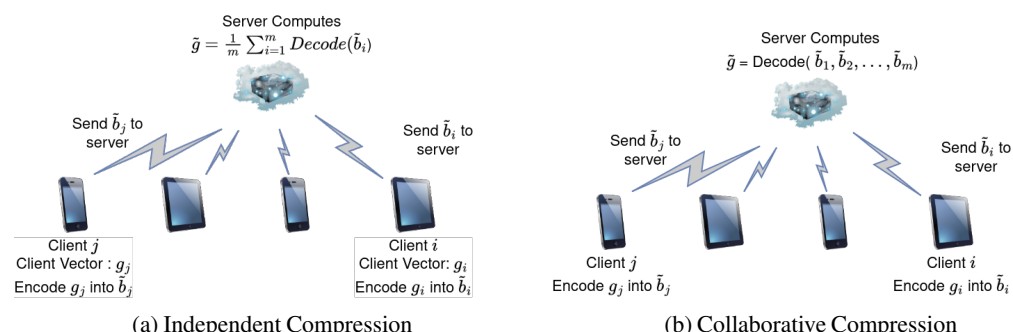

(a) Independent Compression

(b) Collaborative Compression

Figure 1: Compression for Distributed Mean Estimation

The naive strategy of clients sending their vectors $g_i$ to the server for DME incurs no error, however, has a high communication cost, rendering it useless in most of the real-world network applications. A principled way to tackle this is to use compression: each client $i \in [m]$ compresses its vector $g_i$ into an efficient encoding $\tilde{b}_i \in \mathcal{B}_i$ which can then be sent to the server; The server forms an estimate $\tilde{g}$ of the mean $g$ using the encodings $\{\tilde{b}_i\}_{i \in [m]}$. We can then compute the error of the estimate $\tilde{g}$ and the number of bits required to communicate $\tilde{b}_i$ (i.e., $\log_2 |\mathcal{B}_i|$) to analyze the efficiency of the compression scheme. As opposed to distributed statistical inference Braverman et al. (2016); Garg et al. (2014), we do not assume that $g_i$ are sampled from a distribution, and instead the estimation error of these schemes is computed in terms of $g_i$.

One way to approach this compression paradigm is when each client compresses its vector oblivious to others, and the server separately decodes the vectors before aggregating (Figure 1a). We call this *independent compression* and several existing works Konečný & Richtárik (2018); Suresh et al. (2017); Safaryan et al. (2021); Gandikota et al. (2022); Vargaftik et al. (2021) use such a compression scheme. The simplest example of this scheme is RandK Konečný & Richtárik (2018), where each client sends only $K \in \mathbb{N}$ coordinates as $\tilde{b}_i$, and the server estimates $\tilde{g}$ as the average of $K$-sparse vectors from each client. As $K < d$, this scheme requires less communication than sending the full vector $g_i$ from each client $i \in [m]$. Note that independent compressors are a specific class among the more general possible compressors.

However, independent compressors suffer from a significant drawback, especially when the vectors to be aggregated are similar/not-too-far, which is often the case for gradient aggregation in distributed learning. Consider the case when two distinct clients $i, j \in [m]$ have different vectors $g_i \neq g_j$, but they differ in only one coordinate. Then, independent compressors like RandK will end up sending $\tilde{b}_i$ and $\tilde{b}_j$ which are very similar (in fact, same with high probability) to each other, and therefore wasting communication.

Collaborative compressors Suresh et al. (2022); Szlendak et al. (2021); Jhunjhunwala et al. (2021); Jiang et al. (2023) can alleviate this problem. Figure 1b describes a collaborative compressor, where the encodings $\{\tilde{g}_i\}_{i \in [m]}$ may not be independent of each other and a decoding function *jointly* decodes all encodings to obtain the mean estimate $\tilde{g}$. Clearly, this opens up more possibilities to reduce communication - but also the error of collaborative compressors can be made to scale as the variance of the vectors instead of their norms. Whereas, in independent compression a lot of communication is also spent in figuring out their norms separately.

The amount of required communication also depends on the metric for estimation error. Among the existing schemes for collaborative compressors, most provide guarantees on the $\ell_2$ error $||\tilde{g} - g||_2^2$ Suresh et al. (2022); Szlendak et al. (2021); Jhunjhunwala et al. (2021); Jiang et al. (2023). Also, in collaborative compressors, the error must ideally be dependent on *some measure of correlation/distance* among the vectors, which is indeed the case for all of these schemes. In this paper, the measure of such a distance is denoted with $\Delta$, with some subscript signifying the exact measure; the vectors in question have high similarity as $\Delta \to 0$. The estimation error naturally grows with the dimension $d$, and decays with the number of clients $m$ (due to an averaging). One of our major contributions is to design a compression scheme that has significantly improved dependence on the number of clients $m$ to counter the effect of growing dimension $d$.

If one were to estimate the unit vector in the direction of the average vector $\frac{1}{m}\sum_{i=1}^m g_i$, which is often important for gradient descent applications, using an estimate of the mean with low $\ell_2$ error can be

| Compressor | Error metric | Error | # Bits/client |
|---|---|---|---|
| NoisySign (Algorithm 1) | $\|\tilde{g}-g\|_\infty$ | $\left(1-\frac{\Delta_\Phi+\sqrt{\frac{\log m}{m}}(\sqrt{\Delta_\Phi}+\sqrt{\alpha(\|g\|_\infty)})}{\alpha(\|g\|_\infty)}\right)^{-1}-1$ | $d$ |
| HadamardMultiDim (Algorithm 3) | $\mathbb{E}[\|\tilde{g}-g\|_\infty]$ | $\frac{B}{2^{m-1}}+\Delta_{\text{Hadamard}}$ | $d$ |
| SparseReg (Algorithm 4) | $\mathbb{E}[\|\tilde{g}-g\|_2^2]$ | $B^2\exp\left(-\frac{2m\log L}{d}\right)+\Delta_{\text{reg}}$ | $\log L$ ($L\geq 1$ tunable) |
| OneBit (Algorithm 5) | $\arccos\langle\tilde{g},g\rangle$ | $\pi(\Delta_{\text{corr}}+\frac{d}{mt})$ | $t$ ($t\geq 1$ tunable) |

Table 1: Theoretical results for our proposed collaborative compression schemes. $\Delta_\Phi,\Delta_{\text{Hadamard}},\Delta_{\text{reg}}$ and $\Delta_{\text{corr}}$ are measures of average dissimilarity between vectors $\{g_i\}_{i\in[m]}$ defined in Theorems 4, 1, 2 and Lemma 1 respectively. For NoisySign, $\alpha(x)=1-\Phi_\sigma(x)$ for any $x\in\mathbb{R}$, where $\Phi_\sigma(x)=\text{erf}(\frac{t}{\sqrt{2}\sigma})$ with erf being the error function Glaisher (1871) and $\sigma>0$ is an algorithm parameter. For HadamardMultiDim, we assume $\|g_i\|_\infty\leq B,\forall i\in[m]$. For SparseReg, we assume $\|g_i\|_2\leq B,\forall i\in[m]$ and $L$ is an algorithm parameter. For OneBit, $g$ is the unit vector along the average $\frac{1}{m}\sum_{i=1}^m g_i$ and $\tilde{g}$ is also a unit vector.

highly sub-optimal as the $\ell_2$ error might be large even if all the vectors point in the same direction but have different norms. For this the cosine distance $\arccos(\frac{\langle\tilde{g},g\rangle}{\|\tilde{g}\|\|g\|})$ is a better measure, which has not been studied in the literature. We also give a compression scheme specifically tailored for this error metric. Another interesting metric is the $\ell_\infty$-error which has also not been studied except for in Suresh et al. (2022). There as well, we give an improved dependence of the estimation error on $m$.

Further drawback of existing collaborative compressors such as, Jhunjhunwala et al. (2021); Jiang et al. (2023) is that they require the knowledge of correlation between vectors before employing their compression. Without this knowledge, their error guarantees do not hold.

**Notation.** Let $[n]\equiv\{1,2,...,n\}$. We use $g^{(j)}$ to denote the $j^{th}$ coordinate of a vector $g\in\mathbb{R}^d,j\in[d]$. For a permutation $\rho$ on $[m]$, $\rho^{(i)}$ denotes mapping of $i\in[m]$ under $\rho$.

**Our contributions.** We provide four different collaborative compressors, which are communication-efficient, give error guarantees for different error metrics ($\ell_2$ error, $\ell_\infty$ error and cosine distance), and exhibit optimal dependence on the number of clients $m$ and the diameter of ambient space $B$. To see the advantage of collaboration, we define few natural similarity metrics. All our schemes show graceful degradation of error with the similarity metric between different clients. Our schemes have three subroutines: `Init` which corresponds to initial steps, `Encode` which is performed individually at each client to obtain their encoding $\tilde{b}_i$ and `Decode` which is performed at the server on all the encodings to obtain estimate of mean $\tilde{g}$.

We now provide our main contributions. The theoretical guarantees for our algorithms are summarized in Table 1.

1. We provide a simple collaborative scheme based on the popular signSGD Bernstein et al. (2018a) scheme, NoisySign (Algorithm 1), where sign of each coordinate of a vector is sent after adding Gaussian noise. An advantage of this scheme, compared to others is that we can infer the vector $g$ with an $\ell_\infty$ error guarantee increasing with $\|g\|_\infty$ and decreasing with $m$, without the knowledge of $\|g\|_\infty$ itself. The dissimilarity is $\Delta_\Phi=\mathcal{O}(\frac{1}{m\sigma}\sum_{i=1}^m\|g-g_i\|_\infty)$, where $\sigma$ is the variance of the noise added (Theorem 4). The details of this scheme is delegated to Appendix A.

2. ($\ell_\infty$-**guarantee**) For vectors with $\ell_\infty$ norm bounded by $B$, we propose a collaborative compression scheme, HadamardMultiDim (Algorithm 3) which performs coordinate-wise collaborative binary search. We obtain the best dependence on $m$ and $B$ for the $\ell_\infty$ error ($\mathcal{O}(B\cdot\exp(-m))$) while suffering from an extra error term $\Delta_{\text{Hadamard}}$, which is a measure of average dissimilarity between compressed vectors. $\Delta_{\text{Hadamard}}$ lies in the range $[\Delta_\infty,\Delta_{\infty,\max}]$ where $\Delta_\infty=\max_{j\in[d]}\frac{1}{m}\sum_{i=1}^m|g_i^{(j)}-g^{(j)}|$ and $\Delta_{\infty,\max}=\max_{j\in[d],i\in[m]}|g_i^{(j)}-g^{(j)}|$ (Theorem 1). In Section 2.3, we provide a practical example where value of $\Delta_{\text{Hadamard}}$ can be approximated and use it compare theoretical guarantees of HadamardMultiDim with those of baselines in Table 2.

3. ($\ell_2$-**guarantee**) For vectors with $\ell_2$ norm bounded by $B$, we provide a collaborative compression scheme SparseReg (Algorithm 4) based on Sparse Regression Codes Venkataramanan et al. (2014b;a). We obtain the best dependence on $B$ and $m$ for the $\ell_2$ error ($\mathcal{O}(B\exp(-m/d))$) while compressing to much less than $d$ bits (in fact, to a constant number of bits) per client. The error consists of a penalty for the dissimilarity, $\Delta_{\mathrm{reg}}$, the average dissimilarity between compressed vectors which lies in the range $[\Delta_2, \Delta_{2,\max}]$ where $\Delta_2 = \frac{1}{m}\sum_{i=1}^m ||g - g_i||_2^2$ and $\Delta_{2,\max} = \max_{i \in [m]} ||g - g_i||_2^2$ (see, Theorem 2).

4. (**cosine-guarantee**) For unit norm vectors $\{g_i\}_{i \in [m]}$, we estimate the unit vector $g$ in the direction of the average $\frac{1}{m}\sum_{i=1}^m g_i$. For this, motivated by one-bit compressed sensing Boufounos & Baraniuk (2008), our collaborative compression scheme, OneBit (Algorithm 5), sends the sign of the inner product between the vector $g_i$ and a random Gaussian vector. By establishing an equivalence to halfspace learning with malicious noise, we propose two decoding schemes: the first one is based on Shen (2023) which is optimal for halfspace learning but harder to implement and a second one, based on Kalai et al. (2008) which is easy to implement. Both schemes are computationally efficient, and have an extra dissimilarity term in the error, $\Delta_{\mathrm{corr}} = \frac{1}{m\pi}\sum_{i=1}^m \cos^{-1}(\langle g, g_i\rangle)$, which is the appropriate dissimilarity between unit vectors (see Theorem 3).

5. (**Experiments**) We perform a simulation for DME with our schemes as the dissimilarities vary and compare the three different error metrics from above with various existing baselines (Fig 2a-2c). We also used our DME subroutines in the downstream tasks of KMeans, power iteration, and linear regression on real (and federated) datasets (Fig 2d-2i). Our schemes have lowest error in all metrics for low dissimilarity regime.

---

**Algorithm 1** NoisySign

Encode ($g_i$)

Sample $\xi_i \sim \mathcal{N}(0, \sigma^2 \mathbb{I}_d)$
$\tilde{b}_i = \mathrm{sign}(g_i + \xi_i)$
**return** $\tilde{b}_i$.

Decode ($\{\tilde{b}_i\}_{i \in [m]}$)

$\tilde{g}^{(j)} \leftarrow \Phi_\sigma^{-1}(\frac{1}{m}\sum_{i=1}^m \tilde{b}_i^{(j)}), j = 1,...,d$
**return** $\tilde{g}$

---

**Algorithm 2** Hadamard1DEnc

**Input:** Scalar $s$, Level $K$
$S_K^- = \cup_{k=0}^{K-1}[-B + \frac{2kB}{2^{K-1}}, -B + \frac{(2k+1)B}{2^{K-1}}]$
**return** $-1$ if $s \in S_K^-$ else $+1$

---

**Algorithm 3** HadamardMultiDim

Init()

Clients and server share $\rho$, a random permutation on $[m]$.

Encode ($g_i$)

**for** $j \in [d]$ **do**
$\quad \tilde{b}_i^{(j)} \leftarrow$ Hadamard1DEnc($g_i^{(j)}, \rho^{(i)}$)
**end for**
**return** $\tilde{b}_i$

Decode ($\{\tilde{b}_i\}_{i \in [m]}$)

**for** $j \in [d]$ **do**
$\quad \tilde{g}^{(j)} = \sum_{i=1}^m \tilde{b}_i^{(j)} \cdot \frac{B}{2^{\rho^{(i)}-1}}$
**end for**
**return** $\tilde{g}$

---

**Organization.** In the next subsection, we present related works in distributed mean estimation. The NoisySign algorithm is given in Algorithm 1, and its analysis can be found in Appendix A. In Section 2, we present the two schemes obtaining optimal dependence on $m$, HadamardMultiDim in Subsection 2.1 and SparseReg in Subsection 2.2. In Section 3, we analyze the OneBit compression scheme. Finally, in Section 4, we provide experimental results for our schemes.

## 1.1 RELATED WORKS

**Compressors in Distributed Learning.** Starting from Konečnỳ et al. (2016) most compression schemes in distributed learning involve either quantization or sparsification. In quantization schemes, the real valued input space is quantized to specific levels, and each input is mapped to one of these quantization levels. A theoretical analysis for unbiased quantization was provided in Alistarh et al. (2017). Subsequently, the distributed mean estimation problem with limited communication was formulated in Suresh et al. (2017) where two schemes, stochastic rotated quantization (SRQ) and variable length coding, were proposed. These schemes matched the lower bound for communication and $\ell_2$ error in terms of $\tilde{B}^2 = \frac{1}{m}\sum_{i=1}^m ||g_i||_2^2$. Performing a coordinate-wise sign is also a quantization operation, introduced in Bernstein et al. (2018b). Further advances in quantization include multiple quantization

| Compressor | Error | # Bits/client | Notes |
|---|---|---|---|
| RandK Konečný & Richtárik (2018) | $\mathcal{O}(\frac{d}{K}\tilde{B}^2)$ | $32K+K\log d$ | Independent |
| SRQ Suresh et al. (2017) | $\mathcal{O}(\frac{\log d}{m(K-1)^2}\tilde{B}^2)$ | $Kd$ | Independent |
| Kashin Safaryan et al. (2021) | $\mathcal{O}\left(\left(\frac{10\sqrt{\lambda}}{\sqrt{\lambda}-1}\right)^4\tilde{B}^2\right)$ | $31+\lambda d$ | Independent |
| Drive Vargaftik et al. (2021) | $\mathcal{O}(\tilde{B}^2)$ | $32+d$ | Independent |
| PermK Szlendak et al. (2021) | $\mathcal{O}((1-\max\{0,\frac{m-d}{m-1}\})\Delta_2)$ | $32K+K\log d$ | Collaborative |
| RandKSpatial Jhunjhunwala et al. (2021) | $\mathcal{O}(\frac{d}{mK}\Delta_2)$ | $32K+K\log d$ | Needs Correlation |
| RandKSpatialProj Jiang et al. (2023) | $\mathcal{O}(\frac{d}{mK}\Delta_2)$ | $32K+K\log d$ | Needs Correlation |
| Correlated SRQ Suresh et al. (2022) | $\mathcal{O}\left(\frac{1}{m}\min\{\frac{\sqrt{d}\Delta_\infty^d B}{K},\frac{dB^2}{K^2}\}\right)$ | $2d\log K+K\log d$ | $\|g_i\|_2\leq B,\forall i\in[m]$ |

Table 2: Comparison of existing independent and collaborative compressors in terms of $\ell_2$ error and bits communicated. $K$ is the number of coordinates communicated for sparsification methods(RandK, PermK, RandKSpatial, RandKSpatialProj) and the number of quantization levels for quantization methods (SRQ, vqSGD, Correlated SRQ). The constant $\lambda$ is a parameter of the Kashin scheme. Further, $\tilde{B}^2=\frac{1}{m}\sum_{i=1}^m\|g_i\|_2^2, \Delta_2=\frac{1}{m}\sum_{i=1}^m\|g_i-g\|_2^2$, and $\Delta_\infty=\max_{j\in[d]}\frac{1}{m}\sum_{i=1}^m|g_i^{(j)}-g^{(j)}|$. It is also assumed that a real is equivalent to 32 bits, which is an informal norm in this literature.

levels Wen et al. (2017), probabilistic quantization with noise Chen et al. (2020); Jin et al. (2021); Safaryan & Richtarik (2021), vector quantization Gandikota et al. (2022), and applying structured rotation before quantization Vargaftik et al. (2021); Safaryan et al. (2021). Sparsification involves selecting only a subset of coordinates to communicate. Common examples include RandK Konečný & Richtárik (2018), TopK Stich et al. (2018) and their combinations Beznosikov et al. (2022). Note, for all independent compressors, the $\ell_2$ error scales as $\tilde{B}^2$.

**Collaborative Compressors.** PermK Szlendak et al. (2021) was the first collaborative compressor, where each client would send a different set of $K$ coordinates. Their error scales with the empirical variance, $\Delta_2=\frac{1}{m}\sum_{i=1}^m\|g_i-g\|_2^2$. If $\Delta_2$ is known, or one of the vectors $g_i$ is known, the lattice-based quantizer in Davies et al. (2021) and correlated noise based quantizer in Mayekar et al. (2021) obtains $\ell_2$ error in terms of $\Delta_2$. Further, RandKSpatial Jhunjhunwala et al. (2021) and RandKSpatialProj Jiang et al. (2023) utilize the correlation information to obtain the correct normalization coefficients for RandK with rotations, obtaining guarantees in terms of $\Delta_2$. In absence of correlation information, they propose a heuristic. A quantizer also based on correlated noise, was proposed in Suresh et al. (2022) which achieves the lower bound for scalars. However, for $d$-dimensional vectors of $\ell_2$-norm at most $B$, their dependence on dimension $d$ and number of clients $m$ can be improved by our schemes.

We provide a summary of existing compressors in Table 2, along with their error guarantees.

## 2  OPTIMAL DEPENDENCE ON $m$

If $\|g\|_\infty$ or $\|g\|_2$ is bounded, we can obtain an almost optimal exponential decay with $m$. We provide two schemes that obtain optimal $\ell_\infty$ ( by modifying the sign compressor) and $\ell_2$ error dependence in terms of $m$ and the diameter of the space $B$.

### 2.1  HADAMARDMULTIDIM

When the vectors have bounded $\ell_\infty$ norm, instead of obliviously using the sign compressor on every coordinate on every client, one may be able to divide their range and cleverly select bits to encode the most information. We call our algorithm Hadamard scheme, because the binary-search method involved is akin to the rows of a Hadamard-type matrix.

**Assumption 1** (Bounded domain). $\|g_i\|_\infty\leq B,\forall i\in[m]$.

This would imply that for any $j\in[d]$, $g_i^{(j)}\in[-B,B],\forall i\in[m]$. Now, consider the $i^{th}$ client and the scalar $g_i^{(j)}$ and assume that we are allowed to encode this using $m$ bits. The best error that we can achieve is $\frac{B}{2^{m-1}}$, by performing a binary search on the range $[-B,B]$ for $g_i^{(j)}$, sending one bit per level of the binary search. However, this scheme is not collaborative. To obtain a collaborative scheme, for some permutation $\rho$ on the set of clients $[m]$, the $i^{th}$ client can perform binary search until level $\rho^{(i)}$

and sends its decision at level $\rho^{(i)}$. In this case, each client sends only 1 bit per coordinate. To decode $\tilde{g}^{(j)}$, we take a weighted sum of the signs obtained from different clients weighed by their coefficients $\frac{B}{2^{\rho^{(i)}-1}}$. This is the core subroutine (Algorithm 2). The full compression scheme for $d$ coordinates applies this coordinate-wise in Algorithm 3. Note that, the clients and the server should share the permutation $\rho$ before encoding and decoding, which need not change over different instantiations of the mean estimation problem. To understand the core idea of the scheme, consider the case when all vectors $g_i = g$. Then, sending a different level from a different client is equivalent to doing a full binary search to quantize $g$. As long as $g_i$s are close to $g$, we hope that this scheme should give us a good estimate of $g$. Suppose, $\tilde{b}_{i,k}^{(j)}$ denotes the encoding of $g_i^{(j)}$ at level $k$ $\forall i, k \in [m], j \in [d]$.

**Theorem 1** (HadamardMultiDim Error). *Under Assumptions 1, the estimation error for Algorithm 3 is*

$$\mathbb{E}[||\tilde{g}-g||_\infty] \leq \frac{B}{2^{m-1}} + \min\{\Delta_{\text{Hadamard}}, \Delta_{\infty,\max}\}, \tag{2}$$

*where* $\Delta_{\text{Hadamard}} \equiv \max_{r \in [d]} \sqrt{\frac{1}{m^2} \sum\sum_{1 \leq i \neq j \leq m} \sum_{k=1}^{m} \left(\frac{B(\tilde{b}_{i,k}^{(r)} - \tilde{b}_{j,k}^{(r)})}{2^{k-1}}\right)^2}$ , *and* $\Delta_{\infty,\max} \equiv$ $\max_{r \in [d], i \in [m]} |g_i^{(r)} - g^{(r)}|$.

We provide the proof for this theorem in Appendix D.1. The first term corresponds to the error for binary search, and has an exponential decay with number of clients. In contrast, all previous schemes give $\text{poly}(1/m)$ dependence (see, Table 2). The second term is the price we pay for dissimilarity between the vectors. The term $\Delta_{\text{Hadamard}}$ is the average of the pairwise difference between the encodings at each level. As long as vectors $g_i$ and $g_j$ are similar and their encodings do not differ on a lot of levels, $\Delta_{\text{Hadamard}}$ is small. The following is an interpretable bound on $\Delta_{\text{Hadamard}}$.

$$\Delta_{\text{Hadamard}} \geq \frac{1}{\sqrt{3}} \Delta_\infty - \sqrt{\frac{2(m-1)}{m}} \frac{B}{2^{m-1}}, \tag{3}$$

where $\Delta_\infty \equiv \max_{r \in [d]} \frac{1}{m} \sum_{i=1}^{m} |g_i^{(r)} - g^{(r)}|$. The proof of this is provided in Appendix D.2. As we allow full collaboration between clients, in the worst case, we might have to incur a cost $\Delta_{\infty,\max}$ which is the worst case dissimilarity among clients. However, if client vectors are close, we might end up paying a much lower cost.

---

**Algorithm 4** SparseReg

`Init()`
Clients and server share $A \in \mathbb{R}^{mL \times d}$, and $\rho$, a random permutation on $[m]$
`Encode(`$g_i$`)`
$g_i' \leftarrow g_i$
**for** $j \in [\rho^{(i)}]$ **do**
    $\tilde{b}_{i,j} \leftarrow \text{argmax}_{r \in [L]} \langle A_{(j-1)L+r}, g_i' \rangle$
    $g_i' \leftarrow g_i' - c_j A_{(j-1)L + \tilde{b}_{i,j}}$
**end for**
$\tilde{b}_i \leftarrow \tilde{b}_{i,\rho^{(i)}}$
**return** $\tilde{b}_i$
`Decode(`$\{\tilde{b}_i\}_{i \in [m]}$`)`
$\tilde{g} \leftarrow \sum_{i \in [m]} c_{\rho^{(i)}} A_{(\rho^{(i)}-1)L + \tilde{b}_i}$

$$c_i = B\sqrt{\frac{2\log L}{d^2} \left(1 - \frac{2\log L}{d}\right)^{i-1}} \tag{4}$$

**Algorithm 5** OneBit

`Init()`
Clients and server share unit vectors $\{z_i\}_{i \in [m]}$.
`Encode(`$g_i$`)`
$\tilde{b}_i \leftarrow \text{sign}(\langle g_i, z_i \rangle)$
**return** $\tilde{b}_i$
`Decode(`$\{\tilde{b}_i\}_{i \in [m]}$`)`
$g' \leftarrow \begin{cases} \text{(Shen, 2023, Algorithm 1)(Tech. I)} \\ \frac{1}{m} \sum_{i=1}^{m} z_i \tilde{b}_i \text{(Tech. II)} \end{cases}$
$\tilde{g} \leftarrow g' / ||g'||_2$

---

## 2.2 SPARSE REGRESSION CODING

In this part, we extend the coordinate-wise guarantee of the HadamardMultiDim to $\ell_2$ error between $d$-dimensional vectors of bounded $\ell_2$-norm.

**Assumption 2** (Norm Ball). $||g_i||_2 \leq B, \forall i \in [m]$.

To extend the idea of binary search and full collaboration from HadmardMultiDim, we first need a compression scheme which performs binary search on $d$ dimensional vectors with $\ell_2$ error guarantees.

Sparse Regression codes Venkataramanan et al. (2014b;a), which are known to achieve rate-distortion function for a Gaussian source, fit our requirements. Let $A \in \mathbb{R}^{mL \times d}$ for some parameter $L > 0$, where each element of $A$ is sampled iid from $\mathcal{N}(0,1)$ and $A_k$ denotes the $k$th row of $A$. The full algorithm SparseReg is presented in Algorithm 4. To compress a single vector $g$ using $m \log L$ bits, we find the closest vector to $g$ in the first $L$ rows of $A$; say the index of this vector is $\tilde{b}_1$. Similar to binary search, we subtract $c_1 A_{\tilde{b}_1}$ from $g$, where $c_1$ is given in (4) to obtain an updated $g$. We repeat the process using the next set of $L$ rows. Here, each set of $L$ rows corresponds to a single level of binary search, with the coefficients $c_i$ obtained from Eq (4) having a decaying exponent. By carefully selecting the parameters in the proof of (Venkataramanan et al., 2014b, Theorem 1), we can show that this scheme obtains $\ell_2$ error $B \exp(-m)$. We extend this scheme to all clients to allow full collaboration in a manner similar to HadamardMulti-Dim. Each client $i \in [m]$ encodes at level $\rho^{(i)}$ where $\rho$ is a permutation on $[m]$ and the server computes the weighted sum of the encodings from each client with corresponding coefficients $c_{\rho^{(i)}}$.

**Theorem 2** (SparseReg Error). *Under Assumption 2, there exists a matrix $A$ and constants $\delta_1, \delta_2 > 0$, such that the estimation error of Algorithm 4 is*

$$\mathbb{E}_\rho[||g - \tilde{g}||_2^2] \le B^2 (1 + \frac{10 \log L}{d} \exp\left(\frac{m \log L}{d}\right) (\delta_1 + \delta_2))^2 \left(1 - \frac{2 \log L}{d}\right)^m + \min\{\Delta_{\mathrm{reg}}, \Delta_{2,\max}\}$$

$$where, \ \Delta_{\mathrm{reg}} \equiv \frac{1}{m^2} \sum_{i,j \in [m], i \neq j} \sum_{k=1}^m c_k^2 ||A_{(k-1)L + \tilde{b}_{i,k}} - A_{(k-1)L + \tilde{b}_{j,k}}||_2^2, \quad \Delta_{2,\max} \equiv \max_{i \in [m]} ||g - g_i||_2^2.$$

*In fact, a Gaussian matrix $A$ satisfy this with probability $1 - 2m^2 L \exp(-d\delta_1^2/8) - m \left(\frac{L^{2\delta_2}}{\log L}\right)^{-m}$.*

For $d = \Omega(\log m)$, the probability above can be made arbitrarily close to 1 for large $m$. The proof is provided in Appendix D.3. Similar to HadmardMultiDim, the first term has an exponential dependence in $m$ and is obtained from the existing results of Sparse Regression Codes from Venkataramanan et al. (2014b). In terms of $\ell_2$ error this dependence on $m$ is better than all the prior methods.

The dissimilarity term $\Delta_{\mathrm{reg}}$ has a similar structure to $\Delta_{\mathrm{Hadamard}}$ as it is the pairwise difference between encodings of two different vectors at all levels. As long as the vectors are close to each other, this term is not large. Similar to Equation (3), we can interpret $\Delta_{\mathrm{reg}}$ with the following lower bound for Gaussian matrices with the probability given above.

$$\Delta_{\mathrm{reg}} \ge \frac{1}{3} \Delta_2 - 2B^2 \left(1 + \frac{10 \log L}{d} \exp\left(\frac{m \log L}{d}\right) (\delta_1 + \delta_2)\right)^2 \left(1 - \frac{2 \log L}{d}\right)^m, \tag{5}$$

where $\Delta_2 \equiv \frac{1}{m} \sum_{i=1}^m ||g_i - g||_2^2$. The proof of this is provided in Appendix D.4. If the vectors are close to each other we might incur the worst possible error $\Delta_{2,\max}$, but if they are close, we will pay an average price in terms of $\Delta_{\mathrm{reg}}$.

While both the HadmardMultiDim and SparseReg schemes achieve very low communication rate, that comes at the price of $O(m)$ computing in the `Encode` step. This higher cost in computing is to be expected when one wants to exploit the full potential of collaborative compression (e.g., Jiang et al. (2023), where the `Decode` step takes $O(m^2)$ time).

## 2.3 MOTIVATING EXAMPLE

We now provide a example to show that for practical scenarios, the error terms $\Delta_{reg}$ and $\Delta_{\mathrm{Hadamard}}$ are much smaller than their worst case values. Consider the scenario of Theorem 1 ($\ell_\infty$ error) and set $d = 1$. Assume that the first $c$ vectors are $g_1'$ and the remaining $m - c$ vectors are $g_2'$, for some constant $c \ll m$. In this case, $\Delta_{\infty,\max} = (1 - \frac{c}{m}) |g_1' - g_2'| \approx |g_1' - g_2'|$, while $\Delta_\infty \approx \frac{c}{m} |g_1' - g_2'|$. In this scenario, if the compressed values $\tilde{b}$ for $g_1'$ and $g_2'$ according to the HadamardMultiDim differ at $k \in \mathcal{K} \subseteq [m]$ levels, then, $\Delta_{\mathrm{Hadamard}} \approx \sqrt{\frac{c}{m} \sum_{k \in K} (B/2^{k-1})^2} \le \sqrt{\frac{c}{m}} \min_{k \in \mathcal{K}} \frac{B}{2^{k-1}}$. As $\Delta_{\mathrm{Hadamard}}$ averages over all machines, it decreases with $m$ similar to $\Delta_2$ and should be much smaller than $\Delta_{\infty,\max}$. The only case when it is not smaller than $\Delta_{\infty,\max}$ is when $g_1'$ and $g_2'$ are very close, so that $\Delta_{\infty,\max} = \mathcal{O}(\sqrt{m^{-1}})$, but the first level where they differ ($\min_{k \in \mathcal{K}} k$) is very small. One such example is when the quantized values of $g_1'$ in the set $\mathcal{K}$ sorted by the levels in increasing order are $(+1, -1, -1, -1)$ and that of $g_2'$ are $(-1, +1, +1, +1)$. As the vectors are extremely close in this case, the estimation error with $\Delta_{\infty,\max}$

is not very large. Further, if we assume a distributional assumption on the vectors $g_i$, similar to how we generate Figure 2b, obtaining vectors where $\Delta_{\text{Hadamard}} > \Delta_{\infty,\max}$, happens with low probability. Note that a similar example can be constructed for the SparseReg scheme.

We use this example to further compare the error of our proposed schemes to baselines mentioned in Table 2. Consider any $\ell_2$ compressor whose error is either proportional to $\Lambda \tilde{B}^2$ or $\Lambda \Delta_2$ and it sends $\lambda$ bits/client for some $\lambda, \Lambda > 0$. The $\ell_2$ error is defined as $\mathbb{E}[||\tilde{g} - g||_2^2]$ and the $\ell_\infty$ error is defined as $\mathbb{E}[||\tilde{g} - g||_\infty]$, therefore the corresponding $\ell_\infty$ error of these compressors is $\sqrt{\Lambda}\tilde{B}$ or $\sqrt{\Lambda}\Delta_2$. Now, consider the example which we just presented with $d > 1$ and all coordinates being equal for each vector. Therefore, $\Delta_2 \approx \frac{cd}{m}|g_2' - g_1'|^2$, and plugging this in, the $\ell_2$ error of the schemes is $\sqrt{\Lambda}\tilde{B}$ or $\sqrt{\Lambda\frac{cd}{m}}|g_2' - g_1'|$. HadamardMultiDim sends $d$ bits/client, therefore, to compare with any of these schemes, we set $\lambda = d$. For RandK, this would mean setting $K = \frac{d}{32 + \log d}$. Now, if $|g_1'|, |g_2'| \approx B$ but $|g_2' - g_1'| \ll B$, then $\tilde{B} \approx \sqrt{d}B$. Using these approximations, the error of RandK is $\sqrt{(32 + \log d)d}B$, as $\Lambda = 32 + \log d$. This is much larger than the $\ell_\infty$ error of HadamardMultiDim, as the first term is $B \cdot 2^{m-1}$ and the second term $\Delta_{\text{Hadamard}} \approx \sqrt{\frac{c}{m}}|g_2' - g_1'|$. A similar argument holds for all independent compression schemes, as their $\ell_\infty$ error scales as $\tilde{B}$ which in the worst case is $\sqrt{d}B$.

For compressors whose error scales as $\Lambda \Delta_2$ (PermK, RandKSpatial, RandKSpatialProj), by setting $K = \frac{d}{32 + \log d}$, we obtain the same number of bits/client as HadamardMultiDim scheme. Consider RandKSpatialProj, where $\Lambda = \frac{32 + \log d}{m}$, and the error for our example is $\sqrt{c\frac{(32 + \log d)d}{m^2}}|g_2' - g_1'|$. As long as $d > m$, this error is larger than $\Delta_{\text{Hadamard}}$ by constant terms. A similar argument holds for RandKSpatial and PermK. Additionally, note that the theoretical guarantees for RandKSpatial and RandKSpatialProj do not hold if the correlation is not known, as it is required in the algorithm. Without this information, the heuristics they use do not result in theoretical guarantees and their error might become similar to the error of RandK.

The CorrelatedSRQ compressor achieves the lower bound for collaborative compressors for $d = 1$, and is based on a coordinate-wise scheme, hence the $\Delta_\infty$ in its error guarantees. However, for $d \gg 1$, its error scales poorly. For the example described above, $||g_i||_2 \leq \sqrt{d}B$, therefore, the $\ell_\infty$ error for CorrelatedSRQ is $\sqrt{\frac{1}{m}\min\{\frac{d\Delta_\infty^d B}{K}, \frac{d^2 B^2}{K^2}\}}$. Note that even for $K = 2$, correlated SRQ requires double the number of bits/client as HadamardMultiDim. Note that the first term of HadamardMultiDim is $B \cdot 2^{m-1}$ which is much smaller than any of these terms, while $\Delta_{\text{Hadamard}} \approx \sqrt{\frac{m}{c}}\Delta_\infty$ for our example. Therefore, as long as $\left(\frac{m^2 K}{cdB}\right)^{1/(2d-1)} < \Delta_\infty < \frac{\sqrt{cd}B}{mK}$, $\Delta_{\text{Hadamard}}$ is smaller than $\ell_\infty$ error of CorrelatedSRQ. The size of this interval for $\Delta_\infty$ increases as $d$ increases.

With the above example and analysis, we have specified the exact scenarios when HadamardMultiDim outperforms baselines and this can be easily extended to SparseReg.

## 3 ONE-BIT SCHEMES

In this section, our vectors are assumed to belong on the unit sphere $\mathbb{S}^{d-1}$. Further, our goal is to recover the unit vector in the direction of the average vector $g = (\frac{1}{m}\sum_{i \in [m]} g_i)/||\frac{1}{m}\sum_{i \in [m]} g_i||_2$.

**Assumption 3** (Unit vectors). $g_i \in \mathbb{S}^{d-1}, \forall i \in [m]$.

Consider the collaborative compressor where each client has sample $z_i \sim \text{Unif}(\mathbb{S}^{d-1})$ (which are also available to the server apriori). Client $i$ sends the single bit $\tilde{b}_i = \text{sign}(\langle g_i, z_i \rangle)$ to the server. To recover $g$, consider the trivial case when all vectors $g_i$s were equal. Then, each $\tilde{b}_i = \text{sign}(\langle g, z_i \rangle)$, and to recover $g$, the server needs to learn the halfspace corresponding to $g$ from a set of $m$ labeled datapoints. Applying the same method to when $g_i$s are not all the same, we can estimate $g$ by solving the following optimization problem.

$$\min_{\tilde{g} \in \mathbb{S}^{d-1}} \frac{1}{m} \mathbf{1}(\tilde{b}_i \neq \text{sign}(\langle z_i, \tilde{g} \rangle)). \tag{6}$$

Here, $\mathbf{1}(\cdot)$ denotes the indicator function. We can intuitively view (6) as a halfspace learning problem with a groundtruth $g$, but in the presence of noise, as $g_i \neq g$. Learning halfspaces in the presence of

noise is hard in general Guruswami & Raghavendra (2006). In our setting, if we sample $z_i$ from the intersection of the halfspaces with normal vectors $g$ and $g_i$, then the label is $\text{sign}(\langle g, z_i \rangle)$, otherwise, it is $-\text{sign}(\langle g, z_i \rangle)$. We can consider this to be under the malicious noise model, wherein a fraction of datapoints are corrupted.

**Lemma 1** (Malicious Noise)**.** *If $z_i \sim \text{Unif}(\mathbb{S}^{d-1})$ and $\tilde{b}_i = \text{sign}(\langle z_i, g_i \rangle)$, $\forall i \in [m]$, then, with probability $1 - \mathcal{O}(\exp(-m\Delta_{\text{corr}}))$, $\zeta$, the fraction of the set of datapoints $\{(z_i, \tilde{b}_i)\}_{i \in [m]}$ satisfying $\text{sign}(\langle z_i, g_i \rangle) \neq \text{sign}(\langle g, z_i \rangle)$ is equal to $\Theta(\Delta_{\text{corr}})$, where $\Delta_{\text{corr}} \triangleq \frac{1}{m\pi} \sum_{i=1}^{m} \arccos(\langle g_i, g \rangle)$.*

The proof of the lemma is provided in Appendix E.1. Our methods will use $\Delta_{\text{corr}}$ to measure the deviation between clients. For small $\Delta_{\text{corr}}$, we obtain better performance. If $\langle g, g_i \rangle \geq 0, \forall i \in [m]$, then

$$\cos(\pi\Delta_{\text{corr}}) \geq \sqrt{\frac{1}{m} + \frac{2}{m^2} \sum\sum_{1 \leq i < j \leq m} \langle g_i, g_j \rangle}. \tag{7}$$

The proof of the above remark is provided in Appendix E.3.

As long as the corruption level, $\zeta < \frac{1}{2}$, we can hope to recover the halfspace $g$. We provide two techniques – Techniques I and II, to recover $g$, thus yielding two corresponding `Decode` procedures.

The first decoding procedure (Technique I) is a linear time algorithm for halfspace learning in the presence of malicious noise (Shen, 2023, Theorem 3) that provides obtaining optimal sample complexity and noise tolerance.

**Theorem 3** (Error of Technique I)**.** *If $\zeta$ defined in Lemma 1 is less than $\frac{1}{2}$, after running Algorithm 5 with Technique I, with probability $1 - \delta - \mathcal{O}(\exp(-m\Delta_{\text{corr}}))$, we obtain a hyperplane $\tilde{g}$ such that, $\langle \tilde{g}, g \rangle \geq \cos(\pi(\Delta_{\text{corr}} + \frac{d}{m}))$.*

The algorithm itself is fairly complicated. It assigns weights to different points based on how likely they are to be corrupted. The algorithm proceeds in stages, wherein each stage decreases the weights of the corrupted points and solves the weighted version of (6). The key technique is to use matrix multiplicative weights update (MMWU) Arora et al. (2012) to yield linear time implementation of both these steps, instead of Awasthi et al. (2017) which used polynomial time linear programs for this purpose.

Technique II is the simple average algorithm of Servedio (2002), which obtains suboptimal error guarantees. We defer the details of this to Appendix B and the proofs are provided in Appendix E.

## 4 EXPERIMENTS

**Setup.** To compare the performance of our proposed algorithms, we perform DME for three different distributions which correspond to the three error metrics covered by our schemes – $\ell_2, \ell_\infty$ and cosine distance. Then, we run our algorithms as the DME subroutine for three different downstream distributed learning tasks – KMeans, power iteration and linear regression. KMeans and power iteration are run on MNIST LeCun & Cortes (2010) and FEMNIST Caldas et al. (2018) datasets and we report the KMeans cost and top eigenvalue as the metrics. For linear regression, we run gradient descent on UJIndoorLoc Torres-Sospedra et al. (2014) and a Synthetic mixture of regressions dataset, with low dissimilarity between the mixture components, and report the test MSE. We compare against all baselines in Table 2 for 3 random seeds and report the methods which perform the best in Fig 2. Additional details for our experimental setup are deferred to Appendix F.

**Results.** *Distributed Mean Estimation.* From Fig 2a and 2b, HadamardMultiDim and SparseReg, whose error is optimal in $m$, obtain the best performance in terms of $\ell_\infty$ and $\ell_2$ error for low dissimilarity. Especially, for HadamardMultiDim in Fig 2b, the gap in $\ell_\infty$ error to next best scheme is very large. NoisySign obtains competitive performance to other baselines as we use a large $\sigma$. The performance of OneBit for cosine distance metric (Fig 2c) shows that compressors with $\ell_2$ error guarantees perform poorly in terms of cosine distance. For all collaborative compression schemes, including our proposed schemes, performance degrades as dissmilarity increases. From Fig 2a and 2b, the rate of this decrease is more severe for SparseReg than HadamardMultiDim. For large dissimilarity, HadamardMultiDim and SparseReg can perform worse than certain baselines.

*KMeans and Power iteration.* For MNIST dataset, where dissimilarity is low, HadamardMultiDim performs best for KMeans and close to the best baseline for power iteration (Fig 2d and 2e). Most of

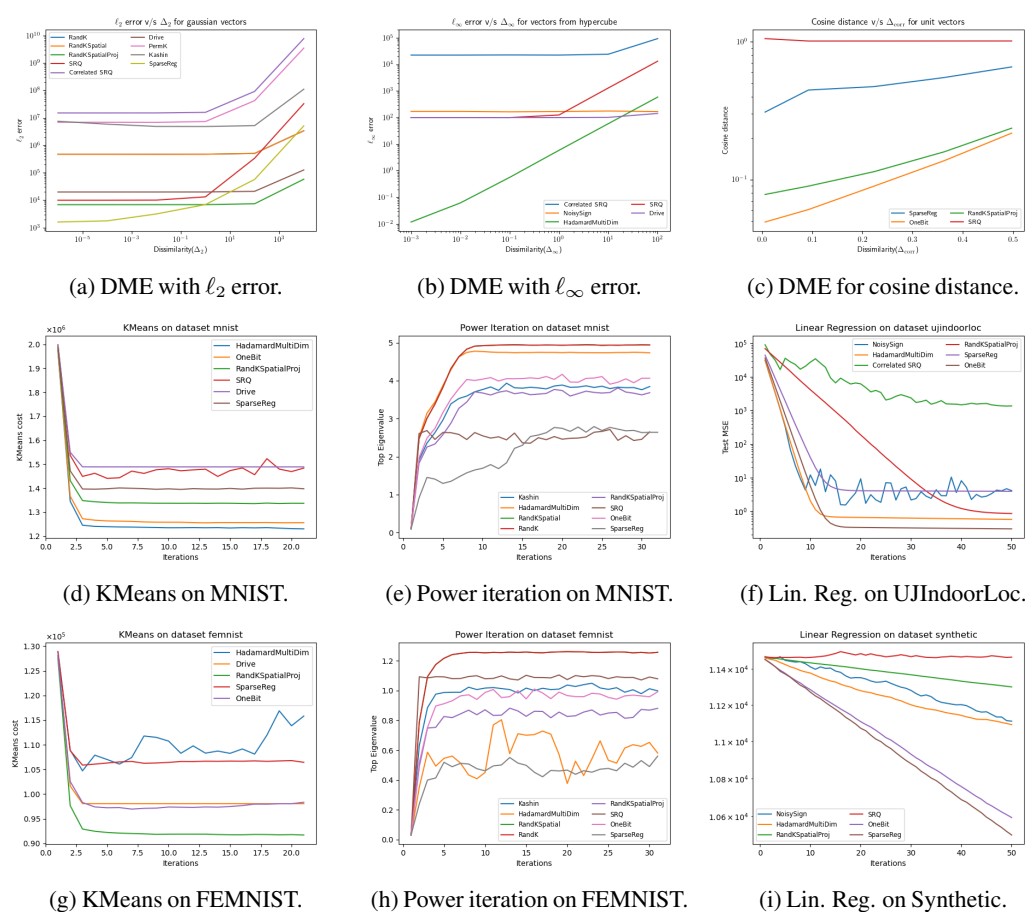

(a) DME with $\ell_2$ error.     (b) DME with $\ell_\infty$ error.     (c) DME for cosine distance.

(d) KMeans on MNIST.     (e) Power iteration on MNIST.     (f) Lin. Reg. on UJIndoorLoc.

(g) KMeans on FEMNIST.     (h) Power iteration on FEMNIST.     (i) Lin. Reg. on Synthetic.

Figure 2: Performance of DME(Distributed Mean Estimation), KMeans, Power iteration and linear regression for the same communication budget. For each experiment, we report the best compressors. Lin. Reg. refer to Linear Regression. For power iteration, higher top eigenvalue is better. For all other experiments, we report the error, so lower is better.

our collaborative compression schemes do not perform as well as RandK on FEMNIST, due to higher client dissimilarity. OneBit is very communication-efficient, so running it for the same communication budget as our baselines ensures that it still remains competitive for KMeans(Fig 2g).

*Linear Regression.* From Fig 2f and2i, all collaborative compressors perform better than independent compressors as UJIndoorLoc and synthetic datasets have low dissimilarity among clients as compared to FEMNIST. Our schemes can take full advantage of this low dissimilarity, so HadamardMultiDim and OneBit outperform baselines on both datasets. As the Synthetic dataset has lower dissimilarity than UJIndoorLoc, even the NoisySign performs better than other baselines, and SparseReg obtains best performance.

## 5 CONCLUSION

We proposed four communication-efficient collaborative compression schemes to obtain error guarantees in $\ell_2$-error (SparseReg), $\ell_\infty$-error (NoisySign, HadamardMultiDim) and cosine distance (OneBitAvg). The estimation error of our schemes improves with number of clients, and degrades with dissimilarity between clients. Our schemes are biased and our dissimilarity metrics ($\Delta_{\text{reg}}$, $\Delta_{\text{Hadamard}}$) depend on the quantization levels. However, these can be improved by using existing techniques for converting biased compressors to unbiased ones Beznosikov et al. (2022) and adding noise before quantization Tang et al. (2023); Chzhen & Schechtman (2023). Lower bounds for collaborative compressors in terms of their dissimilarity metrics will allow us to assess the optimality of our schemes.

Error feedback Karimireddy et al. (2019) reduces the error of independent compressors and it will be interesting to check if it works for our collaborative compressors.

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

## A  NOISYSIGN FOR UNBOUNDED $||g_i||_\infty$

The sign-compressor Bernstein et al. (2018a) applies the sign function coordinate-wise, where $\text{sign}(x) = +1$ if $x \geq 0$ and $-1$ otherwise. For this section, we will focus on a single coordinate $j \in [d]$. Note that for any $i \in [m]$, $\text{sign}(g_i^{(j)})$ does not have information about $|g_i^{(j)}|$. Existing compressors Karimireddy et al. (2020) remedy this by sending $|g_i^{(j)}|$ separately, or assuming that $|g_i^{(j)}|$ is bounded by some constant $B$ Safaryan & Richtarik (2021); Chzhen & Schechtman (2023); Jin et al. (2023); Tang et al. (2023). In the second case, the maximum error that can be incurred is $\frac{B}{2}$. This can be improved by adding uniform symmetric noise before taking signs Chen et al. (2020); Chzhen & Schechtman (2023). However, if no information is available about $|g_i^{(j)}|$, we cannot provide an estimate of $g_i^{(j)}$.

We utilize the concept of adding noise before taking signs, however, to accommodate possibly unbounded $|g_i^{(j)}|$, we add symmetric noise with unbounded support. One choice for such noise is the Gaussian distribution $\mathcal{N}(0, \sigma^2)$. For $\xi_i^{(j)} \sim \mathcal{N}(0, \sigma^2)$, we send $\tilde{b}_i^{(j)} = \text{sign}(g_i^{(j)} + \xi_i^{(j)})$ as the encoding. Note that $\mathbb{E}[\tilde{b}_i^{(j)}] = \Phi_\sigma(g_i^{(j)})$, where $\Phi_\sigma(t) = 2\text{Pr}_{x \sim \mathcal{N}(0,\sigma^2)}[x \geq -t] - 1 = \text{erf}(\frac{t}{\sqrt{2}\sigma})$, and $\text{erf}$ is the error function for the unit normal distribution. A single $\tilde{b}_i^j$ gives us information about $g_i^{(j)}$, however, using it to decode $g_i^{(j)}$ might incur a very large variance. However, assuming that all $g_i^{(j)}$ are close to $g^{(j)}$ for $i \in [m]$, $\frac{1}{m}\sum_{i=1}^m \tilde{b}_i^{(j)}$ is a good estimator for $\Phi_\sigma(g^{(j)})$. So, to estimate $g^{(j)}$, we can use $\Phi_\sigma^{-1}(\frac{1}{m}\sum_{i=1}^m \tilde{b}_i^{(j)})$. This scheme performed coordinate-wise is the NoisySign algorithm described in Algorithm 1.

We provide estimation error for recovering $\tilde{g}$ using this scheme.

**Theorem 4** (Estimation error of noisy sign). *With probability $1 - 2dm^{-c}$, for some constant $c > 0$, the estimation error of Algorithm 1 is*

$$||\tilde{g} - g||_\infty \leq \sqrt{\frac{\pi}{2}}\left(\left(1 - \frac{\Delta_\Phi + \sqrt{\frac{8c\log m}{m}}(\sqrt{\Delta_\Phi} + \sqrt{\alpha(||g||_\infty)})}{\alpha(||g||_\infty)}\right)^{-1} - 1\right), \tag{8}$$

*where $\Delta_\Phi \triangleq \max_{j \in [d]} |\frac{1}{m}\sum_{i=1}^m \Phi_\sigma(g_i^{(j)}) - \Phi_\sigma(g^{(j)})|$ and $\alpha(u) \triangleq 1 - \Phi_\sigma(u)$.*

The proof is provided in Appendix C.1. Applying $\Phi_\sigma^{-1}$ to estimate $g$ makes our scheme collaborative. To gain insight into the error, note that $(1-x)^{-1} - 1 \approx x$, for small $x$. The error increases with the increase in $||g||_\infty$ as we are compressing unbounded variables $g_i$ into the bounded domain $[-1, 1]$ which is the range of the function $\Phi_\sigma$. The number of clients $m$ determines the resolution with which we can measure on this domain, as the value $\frac{1}{m}\sum_{i=1}^m \tilde{b}_i$ can only be in multiples of $\frac{1}{m}$. Therefore, increasing $m$ decreases the error. As $m \to \infty$, the $\ell_\infty$-error approaches $\frac{\Delta_\Phi}{\alpha(||g||_\infty)}$.

Note that $\Delta_\Phi$ determines the average separation between vectors in terms of the $\Phi_\sigma$ operator. If vectors $g_i$ are similar to each other, $\Delta_\Phi$ is small and error is small as a result. Further, $\Delta_\Phi$ can be bounded by more interpretable quantities if the average separation between $g_i$ and $g$ is small:

$$\Delta_\Phi \leq \sqrt{\frac{2}{\pi}}\frac{1}{m\sigma}\sum_{i \in [m]} ||g_i - g||_\infty. \tag{9}$$

Proof of this is provided in Appendix C.2. Note that $\Delta_\Phi$ is always $\leq 1$, so if the average error in terms $\ell_\infty$ norm is much smaller than $\sigma$, then the above bound makes sense. Additionally, one can tune the value of $\sigma$ if additional information about $||g||_\infty$ or $\frac{1}{m}\sum_{i=1}^m ||g_i - g||_\infty$ is known.

Vanilla sign compression without the gradient information will yield a constant error of $\mathcal{O}(\max_{i \in [m]} ||g_i||_\infty)$, as each sign would need to be accurate. However, for large $m$ and small $\Delta_\Phi$ our collaborative compressor performs much better.

# B ANALYSIS OF ONEBIT TECHNIQUE II

**Technique II : Servedio (2002)** (Shen, 2023, Algorithm 1) might be difficult to implement in practice as it involves several subroutines and the knowledge of $\Delta_{\text{corr}}$. Technique II uses the average of the vectors $z_i$ scaled by their signs $\tilde{b}_i$ is used as an estimator for the unit vector $g$

**Theorem 5** (Error of Technique II). *If $\zeta$ defined in in Lemma 1 is less than $\frac{1}{2}$, after running Algorithm 5 with Technique II, with probability $1 - \delta - \mathcal{O}(\exp(-m\Delta_{\text{corr}}))$, we obtain a hyperplane $\tilde{g}$ such that, $\langle \tilde{g}, g \rangle \geq \cos(\pi(\sqrt{d}\Delta_{\text{corr}} + \frac{d}{\sqrt{m}}))$.*

The proofs for Theorems 3 and 5 are provided in Appendix E.2.

The performance of both techniques improves with decrease in $\Delta_{\text{corr}}$. Since we have only $m$ bits to infer a $d$-dimensional vector, we require $m > d$, with Technique II requiring $m > d^2$. If we send $t$ bits per client in OneBit, then the number of samples for the halfspace learning is $mt$, thus obtaining the guarantee in Table 1. The main benefit of OneBit schemes is their extreme communication efficiency.

Existing quantization and sparsification schemes require sending at least $\log K$ or $\log d$, where $K$ is the number of quantization levels.

Note that, we can use compressor for $\ell_2$ error to first decode the mean and then normalize it to obtain its unit vector. If such a scheme uses $t$ bits and has $\ell_2$ error either $\Lambda\Delta_2$ or $\Lambda\tilde{B}^2$ then its cosine similarity $\frac{\langle g,\tilde{g}\rangle}{||g'||_2||\tilde{g}||_2} \geq 1 - \frac{\Lambda}{2||g'||_2^2}$ for $||g'||_2 \approx ||\tilde{g}||_2$, where $g' = \frac{1}{m}\sum_{i=1}^m g_i$ and $\tilde{g}$ is the estimate of $g'$. To compare this with OneBit Technique I, we send $\lambda$ bits per client to obtain the same communication budget. The cosine similarity of this scheme is $\cos(\pi(\Delta_{\text{corr}} + \frac{d}{tm}))$. We can lower bound this similarity by $1 - 2\pi^2\Delta_{\text{corr}}^2 + 2\pi^2\frac{d^2}{m^2t^2}$ as $\cos(x) \geq 1 - \frac{x^2}{2}$. Comparing this cosine similarity with that obtained for $\ell_2$-compressor, as long as $2\pi^2\Delta_{\text{corr}}^2 + 2\pi^2\frac{d^2}{m^2\beta^2} < \Lambda$, OneBit Technique I performs better. For any sparsification scheme sending $K$ coordinates, $\Lambda$ is at least $\frac{d}{mK}$. If we set $t = 32K + K\log d$, OneBit Technique I outperforms the sparsification scheme as long as $\Delta_{\text{corr}}$ is small.

## C PROOFS FOR APPENDIX A

### C.1 PROOF OF THEOREM 4

As all operations are coordinate-wise, we restrict our focus to only a single dimension $j \in [d]$.

$$\mathbb{E}_{\xi_i}[\tilde{b}_i^{(j)}] = \Phi_\sigma(g_i^{(j)}), \forall i \in [m]$$

Note that $\Phi_\sigma(t) = \text{erf}(\frac{t}{\sqrt{2}\sigma})$ and $\Phi_\sigma^{-1}(t) = \sqrt{2}\sigma\,\text{erf}^{-1}(t)$. Further, if $\mathbb{V}ar(\tilde{b}_i^{(j)} - \Phi_\sigma(g_i^{(j)})) = 1 - \Phi_\sigma^2(g_i^{(j)})$. Therefore, by Hoeffding's inequality for random variables with bounded variance, we have,

$$\Pr[|\frac{1}{m}\sum_{i=1}^m(\tilde{b}_i^{(j)} - \Phi_\sigma(g_i^{(j)}))| \geq t] \leq 2\exp\left(-\frac{mt^2}{4(1 - \frac{1}{m}\sum_{i=1}^m\Phi_\sigma^2(g_i^{(j)}))}\right)$$

If we set $t = \sqrt{\frac{4c\log(m)}{m}(1 - \frac{1}{m}\sum_{i=1}^m\Phi_\sigma^2(g_i^{(j)}))}$, for some $c > 0$ in the above inequality, then with probability $1 - 2m^{-c}$, we have,

$$|\frac{1}{m}\sum_{i=1}^m(\tilde{b}_i^{(j)} - \Phi_\sigma(g_i^{(j)}))| \leq t$$

We can represent $\frac{1}{m}\sum_{i=1}^m\tilde{b}_i = \Phi_\sigma(\tilde{g})$, as $\Phi_\sigma$ is an invertible function. To find the difference between $\tilde{g}$ and $g$, we find the difference $\Phi_\sigma(\tilde{g}) - \Phi_\sigma(g)$. With probability $1 - 2m^{-c}$, we have,

$$|\Phi_\sigma(\tilde{g}^{(j)}) - \Phi_\sigma(g^{(j)})| \leq \frac{1}{m}\sum_{i=1}^m|\Phi_\sigma(g_i^{(j)}) - \Phi_\sigma(g^{(j)})| + t$$

To remove the terms of $\Phi_\sigma$, we can apply the function $\Phi_\sigma^{-1}$ on $\tilde{g}^{(j)}$. As $\Phi_\sigma^{-1}$ is not Lipschitz, we need to perform its Taylor's expansion around $\Phi_\sigma(g^{(j)})$ to account for the linear terms in the error. If $\Delta_\Phi = \frac{1}{m}\sum_{i=1}^m|\Phi_\sigma(g_i^{(j)}) - \Phi_\sigma(g^{(j)})|$, then we obtain,

$$|\tilde{g}^{(j)} - g^{(j)}| \leq \max_{u \in [\Phi_\sigma(g^{(j)}) - \Delta_\Phi - t, \Phi_\sigma(g^{(j)}) + \Delta_\Phi + t]}|(\Phi_\sigma^{-1})'(u)|(\Delta_\Phi + t) \tag{10}$$

We now obtain an appropriate upper bound on $(\Phi_\sigma^{-1})'(u)$ as we do not have a closed-form expression for it. We will use the properties of erf to obtain a suitable bound. First, note that $\Phi_\sigma$ and $\Phi_\sigma^{-1}$ are both odd functions, therefore, $|\Phi^{-1}(u)| = |\Phi^{-1}(|u|)|$, so we consider the bound for $u > 0$. Note that

$(\Phi^{-1})'(u) = \frac{1}{\Phi'(\Phi^{-1}(u))}$. For $u > 0$, we have,

$$1 - \text{erf}(u) \leq \exp(-u^2)$$

$$\text{erf}(u) \geq 1 - \exp(-u^2)$$

$$\text{erf}^{-1}(u) \leq \sqrt{-\log(1-u)}$$

$$\Phi_\sigma^{-1}(u) = \sqrt{2}\sigma \text{erf}^{-1}(u) \leq \sigma\sqrt{-2\log(1-u)}$$

$$(\Phi_\sigma^{-1})'(u) = \sqrt{\frac{\pi}{2}}\exp((\Phi_\sigma^{-1}(u))^2/(2\sigma^2)) \leq \sqrt{\frac{\pi}{2}}\exp(-2\log(1-u)/2) = \sqrt{\frac{\pi}{2}}\frac{1}{1-u}$$

For the first step, we use an upper bound on the complementary error function. For the third step, we use the fact that if $f(x) \leq g(x)$, then $f^{-1}(y) \geq g^{-1}(y)$.

Using the following upper bound in Eq (10), we obtain,

$$|\tilde{g}^{(j)} - g^{(j)}| \leq \max_{u \in [\Phi_\sigma(g^{(j)}) - \Delta_\Phi - t, \Phi_\sigma(g^{(j)}) + \Delta_\Phi + t]} \sqrt{\frac{\pi}{2}}\frac{\Delta_\Phi + t}{1 - |u|}$$

$$\leq \sqrt{\frac{\pi}{2}}\frac{\Delta_\Phi + t}{1 - \max\{|\Phi_\sigma(g^{(j)}) - \Delta_\Phi - t|, |\Phi_\sigma(g^{(j)}) + \Delta_\Phi + t|\}}$$

We use $\max\{|\Phi_\sigma(g^{(j)}) - \Delta_\Phi - t|, |\Phi_\sigma(g^{(j)}) + \Delta_\Phi + t|\} \leq \Phi_\sigma(|g^{(j)}|) + \Delta_\Phi + t$, as $\Phi_\sigma$ is an increasing odd function.

$$|\tilde{g}^{(j)} - g^{(j)}| \leq \sqrt{\frac{\pi}{2}}\left(\left(1 - \frac{\Delta_\Phi + t}{1 - \Phi_\sigma(|g^{(j)}|)}\right)^{-1} - 1\right)$$

We first obtain an upper bound for $t$.

$$t = \sqrt{\frac{4c\log m}{m}}\sqrt{1 - \frac{1}{m}\sum_{i=1}^m \Phi_\sigma^2(g_i^{(j)})} = \sqrt{\frac{4c\log m}{m}}\sqrt{1 - \Phi_\sigma^2(g^{(j)}) + \frac{1}{m}\sum_{i=1}^m (\Phi_\sigma^2(g_i^{(j)}) - \Phi_\sigma^2(g^{(j)}))}$$

$$\leq \sqrt{\frac{4c\log m}{m}}\left(\sqrt{1 - \Phi_\sigma^2(g^{(j)})} + \sqrt{\frac{1}{m}|\sum_{i=1}^m (\Phi_\sigma^2(g_i^{(j)}) - \Phi_\sigma^2(g^{(j)}))|}\right)$$

$$\leq \sqrt{\frac{4c\log m}{m}}\left(\sqrt{(1 - \Phi_\sigma(|g^{(j)}|))(1 + \Phi_\sigma(|g^{(j)}|))}\right.$$

$$\left. + \sqrt{|\frac{1}{m}\sum_{i=1}^m (\Phi_\sigma(g_i^{(j)}) - \Phi_\sigma(g^{(j)}))(\Phi_\sigma(g_i^{(j)}) + \Phi_\sigma(g^{(j)}))|}\right)$$

$$\leq \sqrt{\frac{8c\log m}{m}}\left(\sqrt{1 - \Phi_\sigma^2(|g^{(j)}|)} + \sqrt{\Delta_\Phi}\right)$$

We extend the bound to $d$ dimensions by taking a union bound, yielding a probability of error $2dm^{-c}$.

## C.2 PROOF OF EQUATION (9)

The proof follows from using the triangle inequality and a Taylor's expansion for each $\Phi_\sigma(g_i^{(j)})$ around $g^{(j)}$. Note that, for some $u_i^{(j)}$ between $g^{(j)}$ and $g_i^{(j)}$, we have,

$$\Phi_\sigma(g_i^{(j)}) = \Phi_\sigma(g^{(j)}) + \sqrt{\frac{2}{\pi}}\frac{(g^{(j)} - g_i^{(j)})\exp(-\frac{(u_i^{(j)})^2}{2\sigma^2})}{\sigma}$$

$$|\Phi_\sigma(g_i^{(j)}) - \Phi_\sigma(g^{(j)})| \leq \sqrt{\frac{2}{\pi}}\frac{|g^{(j)} - g_i^{(j)}|}{\sigma}$$

We use the fact that $\exp(-\frac{(u_i^{(j)})^2}{2\sigma^2}) \leq 1$. By using triangle inequality for any coordinate $j \in [m]$, we obtain,

$$\Delta_\Phi \leq \max_{j \in [d]} \frac{1}{m} \sum_{i \in [m]} |\Phi_\sigma(g_i^{(j)}) - \Phi_\sigma(g^{(j)})| \leq \frac{1}{m} \sum_{i \in [m]} \max_{j \in [d]} |\Phi_\sigma(g_i^{(j)}) - \Phi_\sigma(g^{(j)})|$$

$$\leq \sqrt{\frac{2}{\pi}} \frac{1}{m} \sum_{i \in [m]} \max_{j \in [d]} \frac{|g^{(j)} - g_i^{(j)}|}{\sigma} \leq \sqrt{\frac{2}{\pi}} \frac{1}{m} \sum_{i \in [m]} \frac{||g - g_i||_\infty}{\sigma}$$

# D    PROOFS OF SECTION 2

## D.1    PROOF OF THEOREM 1

Consider a single dimension $j \in [d]$. Let $g_i^{(j)}$ be the $j^{th}$ coordinate of $g_i$ and $\rho_j$ be the permutation selected for the coordinate $j$. We omit $j$ from $g_i^{(j)}$ and $\rho_j$ to simplify the notation. Let $\tilde{b}_{i,p}$ be the estimate of $g_i$ after decoding it for $p$ levels where $p \in [m]$. Therefore, the estimator $\tilde{g} = \sum_{i=1}^m \frac{\tilde{b}_{i,\rho_i} B}{2^{\rho_i - 1}}$. Let $\tilde{g}_i = \sum_{k=1}^m \frac{\tilde{b}_{i,k} B}{2^{k-1}}$ be the decoded value of $g_i$ till level $m$ and $\bar{g} = \frac{1}{m} \sum_{i=1}^m \tilde{g}_i = \sum_{k=1}^m \frac{\bar{b}_k B}{2^{k-1}}$, where $\bar{b}_k = \frac{1}{m} \sum_{i=1}^m \tilde{b}_{i,k}$.

We compute the expected error for coordinate $j$, where the expectation is wrt the permutation $\rho_j$. Note that $\mathbb{E}_\rho[\tilde{g}_i] = \bar{g}$.

$$\mathbb{E}_\rho[|g - \tilde{g}|] = \sqrt{(\mathbb{E}_\rho[|g - \tilde{g}|])^2} \leq \sqrt{\mathbb{E}_\rho|g - \tilde{g}|^2} \leq \sqrt{\mathbb{E}_\rho|\tilde{g} - \bar{g}|^2 + |g - \bar{g}|^2}$$

$$\leq \sqrt{\mathbb{E}_\rho|\tilde{g} - \bar{g}|^2} + |g - \bar{g}| \leq \frac{1}{m} \sum_{i=1}^m |g_i - \tilde{g}_i| + \sqrt{\mathbb{E}_\rho|\tilde{g} - \bar{g}|^2}$$

$$\leq \frac{B}{2^{m-1}} + \sqrt{\mathbb{E}_\rho|\tilde{g} - \bar{g}|^2}$$

We use Jensen's inequality for the first inequality. For the second inequality, we use bias-variance decomposition for the random variable $\tilde{g}$, where the first term is its variance, and the second term is its bias wrt the term $g$. We then use $\sqrt{a+b} \leq \sqrt{a} + \sqrt{b}$ for any $a, b \geq 0$. To handle the term $|g - \bar{g}|$, we expand both terms as a summation over $m$ clients, followed by a triangle inequality. As each estimator $\tilde{g}_i$ is at least $\frac{B}{2^{m-1}}$ away from $g_i$, each term in the difference $|g_i - \tilde{g}_i|$ has the upperbound $\frac{B}{2^{m-1}}$.

We now bound the variance term separately. Note that

$$\mathbb{E}_\rho|\tilde{g} - \bar{g}|^2 = \mathbb{E}_\rho|\tilde{g}|^2 - \bar{g}^2$$

We first evaluate the second moment $\mathbb{E}_\rho |\tilde{g}|^2$.

$$
\begin{aligned}
\mathbb{E}_\rho |\tilde{g}|^2 &= \mathbb{E}_\rho \left| \sum_{i=1}^m \frac{\tilde{b}_{i,\rho_i}}{2^{\rho_i-1}} \right|^2 = \sum_{i=1}^m \mathbb{E}_\rho \left[ \frac{\tilde{b}_{i,\rho_i}^2 ] B^2}{2^{2\rho_i-2}} \right] + B^2 \sum\sum_{1 \le i \ne j \le m} \mathbb{E}_\rho \left[ \frac{\tilde{b}_{i,\rho_i}}{2^{\rho_i-1}} \frac{\tilde{b}_{j,\rho_j}}{2^{\rho_j-1}} \right] \\
&= \sum_{k=1}^m \frac{B^2}{2^{2k-2}} + B^2 \sum\sum_{1 \le i \ne j \le m} \mathbb{E}_{\rho_i} \left[ \mathbb{E}_\rho \left[ \frac{\tilde{b}_{i,\rho_i}}{2^{\rho_i-1}} \frac{\tilde{b}_{l,\rho_j}}{2^{\rho_j-1}} | \rho_i \right] \right] \\
&= \sum_{k=1}^m \frac{B^2}{2^{2k-2}} + B^2 \sum\sum_{1 \le i \ne j \le m} \mathbb{E}_{\rho_i} \left[ \frac{\tilde{b}_{i,\rho_i}}{2^{\rho_i-1}} \frac{1}{m-1} \sum_{l=1,l \ne \rho_i}^m \frac{\tilde{b}_{j,l}}{2^{l-1}} \right] \\
&= \sum_{k=1}^m \frac{B^2}{2^{2k-2}} + \frac{B^2}{m(m-1)} \sum\sum_{1 \le i \ne j \le m} \sum_{k=1}^m \left[ \frac{\tilde{b}_{i,k}}{2^{k-1}} \sum_{l=1,l \ne k}^m \frac{\tilde{b}_{j,l}}{2^{l-1}} \right] \\
&= \sum_{k=1}^m \frac{B^2}{2^{2k-2}} + \frac{1}{m(m-1)} \sum\sum_{1 \le i \ne j \le m} \left( \sum_{k=1}^m \frac{\tilde{b}_{i,k} B}{2^{k-1}} \right) \left( \sum_{l=1}^m \frac{\tilde{b}_{j,l} B}{2^{l-1}} \right) \\
&\quad - \frac{1}{m(m-1)} \sum\sum_{1 \le i \ne j \le m} \sum_{k=1}^m \frac{B^2 \tilde{b}_{i,k} \tilde{b}_{j,k}}{2^{2k-2}} \\
&= \sum_{k=1}^m \frac{B^2}{2^{2k-2}} + \frac{1}{m(m-1)} \sum\sum_{1 \le i \ne j \le m} \tilde{g}_i \tilde{g}_j - \frac{1}{m(m-1)} \sum\sum_{1 \le i \ne j \le m} \sum_{k=1}^m \frac{B^2 \tilde{b}_{i,k} \tilde{b}_{j,k}}{2^{2k-2}} \\
&= \frac{m^2 |\bar{g}|^2 - \sum_{i=1}^m |\tilde{g}_i|^2}{m(m-1)} + \frac{1}{m(m-1)} \sum\sum_{1 \le i \ne j \le m} \sum_{k=1}^m \frac{B^2 (|\tilde{b}_{i,k}|^2 + |\tilde{b}_{j,k}|^2 - 2\tilde{b}_{i,k}\tilde{b}_{j,k})}{2^{2k-1}} \\
&= \frac{m}{m-1} |\bar{g}|^2 - \frac{\sum_{i=1}^m |\tilde{g}_i|^2}{m(m-1)} + \frac{1}{2m(m-1)} \sum\sum_{1 \le i \ne j \le m} \sum_{k=1}^m \left( \frac{B(\tilde{b}_{i,k} - \tilde{b}_{j,k})}{2^{k-1}} \right)^2
\end{aligned}
$$

Note that we expand the square of the sum of terms where $\tilde{b}_{i,j}^2 = 1$. For the second term, we use the law of total expectation by conditioning on the value of $\rho_i$. To evaluate the inner expectation, we note that $\rho_j$ can take any value other than that of $\rho_i$ with equal probability. To evaluate the outer expectation, note that $\rho_i$ can take any value in $[m]$ with equal probability. In the fourth equation, we subtract the term where $l = k$. Then, we can factorize the remaining terms to obtain $\tilde{g}_i$ and $\tilde{g}_j$. Note that the sum of the product terms $\tilde{g}_i \tilde{g}_j$ can be expressed as $|\sum_{i=1}^m \tilde{g}_i|^2$, with the square terms subtracted. Further, we express the term $\frac{B^2}{2^{2k-2}} = \sum\sum_{1 \le i \ne j \le m} \frac{B^2(|\tilde{b}_{i,k}|^2 + |\tilde{b}_{j,k}|^2)}{2^{2k-1}}$ as $|\tilde{b}_{i,k}|^2 = 1$. Finally, we complete the squares for each term $k$.

Using the above value of second moment $\mathbb{E}_\rho |\tilde{g}|^2$, we can compute the variance,

$$
\begin{aligned}
\mathbb{E}_\rho |\tilde{g} - \bar{g}|^2 &= \mathbb{E}_\rho |\tilde{g}|^2 - |\bar{g}|^2 = \frac{|\bar{g}|^2 - \frac{1}{m} \sum_{i=1}^m |\tilde{g}_i|^2}{m-1} + \frac{1}{2m(m-1)} \sum\sum_{1 \le i \ne j \le m} \sum_{k=1}^m \left( \frac{B(\tilde{b}_{i,k} - \tilde{b}_{j,k})}{2^{k-1}} \right)^2 \\
&= \frac{1}{2m^2} \sum\sum_{1 \le i \ne j \le m} \sum_{k=1}^m \left( \frac{B(\tilde{b}_{i,k} - \tilde{b}_{j,k})}{2^{k-1}} \right)^2
\end{aligned}
$$

We use $\bar{g}^2 \le \frac{1}{m} \sum_{i=1}^m |\tilde{g}_i|^2 = \frac{1}{2m^2} \sum\sum_{1 \le i \ne j \le m} (\tilde{g}_i - \tilde{g}_j)^2 \ge \frac{1}{2m^2} \sum\sum_{1 \le i \ne j \le m} \sum_{k=1}^m \left( \frac{B(\tilde{b}_{i,k} - \tilde{b}_{j,k})}{2^{k-1}} \right)^2$.

To simplify this bound, we need to incorporate difference in the actual gradient vectors. For this purpose, we try to bound the differences $|\tilde{b}_{i,k} - \tilde{b}_{j,k}|$ in terms of $\Delta_{ij} \triangleq |g_i - g_i|$. If

Note that if $\Delta_{ij} = |g_i - g_j|$, then $\tilde{b}_{i,k} = \tilde{b}_{j,k}, \forall k \ge \log \left( \frac{B}{\Delta_{ij}} \right)$

## D.2 PROOF FOR EQUATION (3)

For this section, we consider a single coordinate $r \in [d]$.

$$\frac{1}{m}\sum_{i=1}^{m}|g_i^{(r)}-g^{(r)}| = \sqrt{\left(\frac{1}{m}\sum_{i=1}^{m}|g_i^{(r)}-g^{(r)}|\right)^2} \leq \sqrt{\frac{1}{m}\sum_{i=1}^{m}(g_i^{(r)}-g^{(r)})^2}$$

$$\leq \sqrt{\frac{1}{m}\sum_{i=1}^{m}\left(\frac{1}{m}\sum_{j=1,j\neq i}^{m}(g_i^{(r)}-g_j^{(r)})\right)^2} \leq \sqrt{\frac{1}{m^2}\sum\sum_{1\leq i\neq j\leq m}(g_i^{(r)}-g_j^{(r)})^2}$$

$$\leq \sqrt{\frac{3}{m^2}\sum\sum_{1\leq i\neq j\leq m}(\tilde{g}_i^{(r)}-\tilde{g}_j^{(r)})^2 + \frac{6(m-1)}{m^2}\sum_{i=1}^{m}(g_i^{(r)}-\tilde{g}_i^{(r)})^2}$$

$$\leq \sqrt{\frac{3}{m^2}\sum\sum_{1\leq i\neq j\leq m}(\tilde{g}_i^{(r)}-\tilde{g}_j^{(r)})^2 + \frac{6(m-1)}{m}\frac{B^2}{2^{2m-2}}}$$

$$\max_{r\in[d]}\frac{1}{m}\sum_{i=1}^{m}|g_i^{(r)}-g^{(r)}| \leq \sqrt{3}\Delta_{\text{Hadamard}} + \sqrt{\frac{6(m-1)}{m}}\frac{B}{2^{m-1}}$$

$$\Delta_{\text{Hadamard}} \geq \frac{1}{\sqrt{3}}\max_{r\in[d]}\frac{1}{m}\sum_{i=1}^{m}|g_i^{(r)}-g^{(r)}| - \sqrt{\frac{2(m-1)}{m}}\frac{B}{2^{m-1}}$$

For the first inequality, we use $(\sum_{i=1}^{m}a_i)^2 \leq m\sum_{i=1}^{m}a_i^2, \forall a_i \in \mathbb{R}, i \in [m]$. For the second line, we write down the definition of $g^{(r)}$, and use the above identity again. We then add and subtract $\tilde{g}_i^{(r)}$ and $\tilde{g}_j^{(r)}$ and separate the square terms. For each pair $i,j$, we get two terms $(g_i^{(r)}-\tilde{g}_i^{(r)})^2$ and $(g_j^{(r)}-\tilde{g}_j^{(r)})^2$. By summing them up, we get the coefficient of $6(m-1)$. Since $|g_j^{(r)}-\tilde{g}_j^{(r)}| \leq \frac{B}{2^{m-1}}$, and $\sqrt{a+b} \leq \sqrt{a}+\sqrt{b}, \forall a,b > 0$, we get the fourth line. Finally, we take a max over the coordinates $r \in [d]$ to get the term $\Delta_{\text{Hadamard}}$.

## D.3 PROOF FOR THEOREM 2

To obtain the coefficients $c_i$, we replace set $L=m, n=d, R=\log L$ and $\sigma^2 = \frac{B^2}{d}$ in (Venkataramanan et al., 2014a, Eq 2). The proof of this Theorem is same as Theorem 1 for a single dimension, with the coefficients $\frac{B}{2^{j-1}}$ replaced by $c_j$ and $\tilde{b}_{i,k}^{(r)}$ replaced by $A_{(k-1)L+\tilde{b}_{i,k}}$. Following Appendix D.2, we can write down the $\ell_2$ error.

$$\mathbb{E}_{\rho}[||\tilde{g}-g||_2^2] = \mathbb{E}_{\rho}[||g-\mathbb{E}_{\rho}[\tilde{g}]||_2^2] + \mathbb{E}_{pi}[||\tilde{g}-\mathbb{E}_{\rho}[\tilde{g}]||_2^2]$$

$\mathbb{E}[\tilde{g}] = \bar{g} = \frac{1}{m}\sum_{i=1}^{m}\bar{g}_i$, where $\bar{g}_i = \sum_{j=1}^{m}c_jA_{(j-1)L+\tilde{b}_{i,j}}$. By triangle inequality, the first term is $\frac{1}{m}\sum_{i=1}^{m}||g_i-\bar{g}_i||_2^2$, which is bounded individually by $B^2(1 + \frac{10\log L}{d}\exp\left(\frac{m\log L}{d}\right)(\delta_1 + \delta_2))^2\left(1-\frac{2\log L}{d}\right)^m$ by setting $L=m, n=d, R=\log L, \sigma^2 = \frac{B^2}{d}$ and $\delta_0 = 0$ in (Venkataramanan et al., 2014a, Theorem 1).

For the second term, we need to bound $\mathbb{E}[||\tilde{g}||_2^2]$.

$$\mathbb{E}[||\tilde{g}||_2^2] = \frac{1}{m}\sum_{i=1}^{m}\sum_{j=1}^{m}c_i^2||A_{(j-1)L+\tilde{b}_{i,j}}||_2^2$$

$$+ \sum\sum_{1\leq i\neq j\leq m}\mathbb{E}_\rho\left[c_{\pi(i)}c_{\pi(j)}\langle A_{(\pi(i)-1)L+\tilde{b}_{i,\pi(i)}}, A_{(\pi(j)-1)L+\tilde{b}_{j,\pi(j)}}\rangle\right]$$

$$= \frac{1}{m}\sum_{i=1}^{m}\sum_{j=1}^{m}c_i^2||A_{(j-1)L+\tilde{b}_{i,j}}||_2^2$$

$$+ \frac{1}{m(m-1)}\sum\sum_{1\leq i\neq j\leq m}\mathbb{E}_\rho\left[c_{\pi(i)}c_{\pi(j)}\langle A_{(\pi(i)-1)L+\tilde{b}_{i,\pi(i)}}, A_{(\pi(j)-1)L+\tilde{b}_{j,\pi(j)}}\rangle\right]$$

$$= \frac{m^2||\bar{g}||_2^2 - \sum_{i=1}^{m}||\tilde{g}_i||_2^2}{m(m-1)} + \frac{1}{m(m-1)}\sum\sum_{1\leq i\neq j\leq m}\sum_{k=1}^{m}c_k^2||A_{(k-1)L+\tilde{b}_{j,k}} - A_{(k-1)L+\tilde{b}_{i,k}}||_2^2$$

The remainder of the proof follows proof of Theorem 1 with $|\cdot|^2$ replaced by $||\cdot||_2^2$.

## D.4 PROOF OF EQ (5)

The proof follows that of Eq (3) from Appendix D.2.

$$\Delta_2 = \frac{1}{m}\sum_{i=1}^{m}||g_i-g||_2^2 \leq \frac{1}{m^2}\sum\sum_{1\leq i\neq j\leq m}||g_i-g_j||_2^2$$

$$\leq \sqrt{\frac{3}{m^2}\sum\sum_{1\leq i\neq j\leq m}||\tilde{g}_i-\tilde{g}_j||_2^2 + \frac{6(m-1)}{m^2}\sum_{i=1}^{m}||g_i-\tilde{g}_i||_2^2}$$

$$\leq 3\Delta_{\text{reg}} + 6B^2(1+\frac{10\log L}{d}\exp\left(\frac{m\log L}{d}\right)(\delta_1+\delta_2))^2\left(1-\frac{2\log L}{d}\right)^m$$

# E PROOFS FOR SECTION 3 AND APPENDIX B

## E.1 PROOF OF LEMMA 1

To prove this Lemma, note that $\tilde{b}_i = sign(\langle g_i, z_i\rangle) \neq sign(\langle g, z_i\rangle)$ only if $z_i$ is sampled from the symmetric difference of $g_i$ and $g$. The probability that a $z_i$ sampled uniformly from $\mathbb{S}^{d-1}$ lies in this symmteric difference is given by $\arccos(\langle g,g_i\rangle)/\pi$. If we set $\Delta_{\text{corr}} = \frac{1}{m\pi}\sum_{i\in[m]}\arccos(\langle g,g_i\rangle)$

Let $\zeta$ be the fraction of $z_i$ such that $\tilde{b}_i \neq sign(\langle g,z_i\rangle)$. Then, by Chernoff bound, we have,

$$\Pr[\zeta \geq (1+\gamma)\Delta_{\text{corr}}] \leq \exp(-\frac{\gamma^2 m\Delta_{\text{corr}}}{2+\gamma})$$

By setting $\gamma$ to be any small constant, we obtain, with probability $1-\mathcal{O}(\exp(-m\Delta_{\text{corr}}))$, atmost $\zeta = \Theta(\Delta_{\text{corr}})$ fraction of datapoints are not generated from the halfspace with normal $g$ and are thus corrupted.

## E.2 PROOFS OF THEOREM 3 AND 5

To prove Theorem 3, we utilize the guarantees of (Awasthi et al., 2017, Theorem 1), where the sample complexity requirement ensures that the error is $\tilde{O}(\frac{d}{m})$. Further, (Awasthi et al., 2017, Theorem 1) obtains error guarantee linear in the noise rate of the samples which is obtained from Lemma 1. The error guarantee is in terms of the symmetric difference between $\tilde{g}$ and $g$ wrt the uniform distribution on the unit sphere. Since this is equal to the angle between these two vectors divided by $\pi$, this gives us a bound on the inner product of these two unit vectors.

To prove Theorem 5, from (Kalai et al., 2008, Theorem 12), the sample complexity provides the term $\frac{d}{\sqrt{m}}$ while the noise tolerance provides the term $\sqrt{d}\Delta_{\text{corr}}$.

### E.3 PROOF OF EQUATION (7)

To prove this remark, note that $\arccos(x)$ is concave for $x \geq 0$. Therefore, by applying Jensen's inequality, we obtain,

$$\Delta_{\text{corr}} = \frac{1}{m\pi}\sum_{i\in[m]}\arccos(\langle g_i, g\rangle) \leq \frac{1}{\pi}\arccos\left(\langle\frac{1}{m}\sum_{i=1}^{m}g_i, g\rangle\right) = \frac{1}{\pi}\arccos\left(||\frac{1}{m}\sum_{i=1}^{m}g_i||_2\langle g, g\rangle\right)$$

$$\leq \frac{1}{\pi}\arccos\left(\sqrt{||\frac{1}{m}\sum_{i\in[m]}g_i||_2^2}\right) = \frac{1}{\pi}\arccos\left(\sqrt{||\frac{\sum_{i\in[m]}\langle g_i, g_i\rangle}{m^2} + \frac{2}{m^2}\sum\sum_{1\leq i<j\leq m}\langle g_i, g_j\rangle||}\right)$$

$$= \frac{1}{\pi}\arccos\left(\sqrt{\frac{1}{m} + \frac{2}{m^2}\sum\sum_{1\leq i<j\leq m}\langle g_i, g_j\rangle}\right)$$

## F  ADDITIONAL EXPERIMENT DETAILS

**Baselines** We implement all the baselines mentioned in Table 2. As all these baselines are suited to $\ell_2$ error, for the DME experiment on gaussians, where $\ell_2$ error is the correct metric, compare SparseReg (Algorithm 4) to all these baselines. For $\ell_\infty$ error uniform distribution, we implement NoisySign (Algorithm 1) and HadamardMultiDim (Algorithm 3) and compare it to Correlated SRQ Suresh et al. (2022), as it's guarantees hold in single dimensions. We also add comparisons to its independent variant, SRQ Suresh et al. (2017), and Drive Vargaftik et al. (2021), which performs coordinate-wise signs. For the unit vector case, we implement OneBit (Algorithm 5 Technique II) and SparseReg(Algorithm 4) and compare it with one independent compressor (SRQ Suresh et al. (2017)) and one collaborative compressor (RandKSpatialProj Jiang et al. (2023)). Note that we set $d=512$ throughout our experiments and tune the parameters (number of coordinates sent Konečný & Richtárik (2018); Jhunjhunwala et al. (2021) or the quantization levels in Suresh et al. (2017; 2022)) so that all compressors have the same number of bits communicated. For compressors without tunable parameters, we repeat them to match the communication budget.

**Datasets** For the distributed mean estimation task, we generate $d$ dimensional vectors on $m=100$ clients. To compare $\ell_2$ error, we generate $g$ with $||g||_2=100$. Then, each client generates $g_i$ from a $\mathcal{N}(0,\Delta_2^2)$, where $\Delta_2 \in [0.001,100]$. To compare $\ell_\infty$ error, we generate $g$ uniformly from a hypercube $[-B,B]^d$ where $B=100$. Each client generates $g_i$ from a smaller hypercube $[-\Delta_\infty,\Delta_\infty]^d$ centered at $g$ where $\Delta_\infty \in [10^{-3},10^2]$. To compare cosine distance, we generate $g$ uniformly from the unit sphere, and each client generates $g_i$ uniformly from the set of unit vectors at a cosine distance $\Delta_{corr}$ from the $g$, Here, $\Delta_{corr} \in [0.01,0.4]$.

For KMeans and power iteration, we set $m=50$. FEMNIST is a real federated dataset where each client has handwritten digits from a different person. We apply dimensionality reduction to set $d=512$. We run 20 iterations of Lloyd's algorithm Lloyd (1982) for KMeans and 30 power iterations. For distributed linear regression, the Synthetic dataset is a mixture of linear regressions, with one mixture component per client. The true model $w_i \in \mathbb{R}^d$ for each component is obtained from DME setup for gaussians with $\Delta_2=4$. Then, we generate $n=1000$ datapoints on each client, where the features $x$ are sampled from standard normal, while the labels $y$ are generated as $y=\langle w_i,x\rangle+\xi$, where $\xi$ is the zero-mean gaussian noise with variance $10^{-2}$. For UJIndoorLoc, we use the first $d=512$ of the 520 features following Jiang et al. (2023). The task for UJIndoorLoc dataset is to predict the longitude of a phone call. For both the linear regression datasets, we run 50 iterations of GD. For MNIST and UJIndoorLoc, we split the dataset uniformly into $m$ chunks one per client.

**Metrics** With the same number of bits, we can directly compare the error of baselines. For mean estimation, we measure $\ell_2$ error, $\ell_\infty$ error and cosine distance for gaussian, uniform and unit vectors respectively. For KMeans, we report the KMeans objective. For power iteration, we report the top eigenvalue. For linear regression, we provide the mean squared error on a test dataset. All the experiments for distributed learning are provided in Figure 2 for the best compressors. For all experiments except power iteration, lower implies better performance. For power iteration, higher implies better performance, as we need to find the eigenvector corresponding to the top eigenvalue.

We provide the code in the supplementary material and all the experiments took $5$ days to run on a single $20$ core machine with $25$ GB RAM.

### F.1 LOGISTIC REGRESSION

In this section, we perform additional experiments to compare our methods to logistic regression on the HAR dataset Reyes-Ortiz et al. (2012). The HAR dataset has $6$ classes of which we select the last two and label them with $\pm 1$. This converts the dataset into a binary classification problem. We split the dataset into $m=20$ clients iid. HAR dataset has $561$ features which we reduce by PCA to $d=512$. We perform logistic regression on this dataset, where the logistic loss for any data point $(x,y)\in\mathbb{R}^d\times\{\pm 1\}$ is defined as $\ell(w,(x,y))=\log(1+\exp(-\langle w,x\rangle\cdot y)$ for any weight $w\in\mathbb{R}^d$. We report the training loss and test accuracy for different baselines after running distributed Gradient Descent with learning rate $0.001$ for $T=200$ iterations in Figure 3. Following earlier plots, we report the best-performing compressors in the plot.

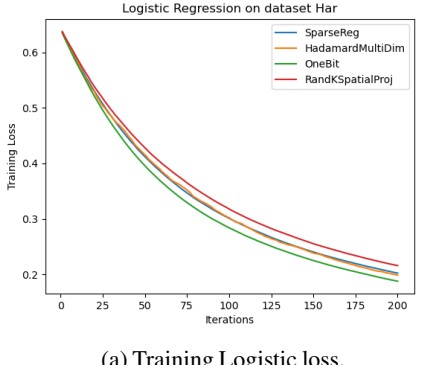
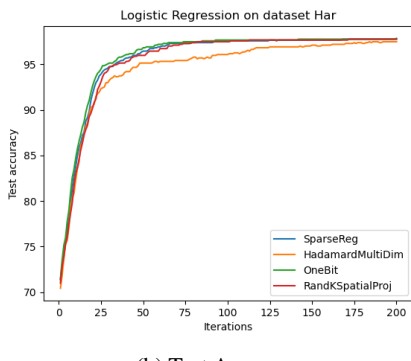

(a) Training Logistic loss.
(b) Test Accuracy.

Figure 3: Performance of compressors for Logistic regression on HAR Reyes-Ortiz et al. (2012) dataset

From the above figure, the best, second best and fourth best compressors in terms of training loss and test accuracy are our compressors, OneBit, SparseReg and HadamardMultDim respectively. Further, among the top $4$ best-performing schemes only one baseline, RandKSpatialProj, comes in the third. This shows the benefit of using collaborative compressors.

## G  DISTRIBUTED GRADIENT DESCENT WITH SPARSEREG COMPRESSOR

This section uses our $\ell_2$ compressor, SparseReg, for running FedAvg. Each client $i\in[m]$ contains a local objective function $f_i:\mathcal{W}\to\mathbb{R}$. We define the global objective function $f(w)=\frac{1}{m}\sum_{i=1}^m f_i(w),\forall w\in\mathcal{W}\subset\mathbb{R}^d$. The goal is to find $w^\star\in\operatorname{argmin}_{w\in\mathcal{W}}f(w)$. Note that $\nabla f(w)=\frac{1}{m}\sum_{i=1}^m\nabla f_i(w)$, therefore, in our case, the vector $g_i$ correspond to $\nabla f_i(w)$. We describe the algorithm in Algorithm 6

We first state the assumptions required for applying the SparseReg compressor.

**Assumption 4** (Bounded Gradient). *For all $w\in\mathcal{W},i\in[m]$, we assume that $||\nabla f_i(w)||_2\leq B$.*

By this assumption, we ensure that for each iteration $t$ in Algorithm 6, $||g_i||_2=||\nabla f_i(w^t)||_2$ is bounded. Further, bounded gradients imply that each $f_i$ is Lipschitz. By triangle inequality, we can also establish the following corollary.

**Corollary 1.** *The objective function $f(w)$ is B-Lipschitz, $\forall w\in\mathcal{W}$.*

From the above assumptions, it is clear that local objective functions need to be Lipschitz. From (Bubeck, 2015, Theorem 3.2), if the domain of iterates, $\mathcal{W}$ is bounded and $f(w)$ is also convex, then gradient descent can converge at a rate $\mathcal{O}(1/\sqrt{T})$. We use these two assumptions, and establish a $\mathcal{O}(1/\sqrt{T})$ rate along with a error obtained from Theorem 2. We define $\Delta_{\mathrm{reg}}(t)$ and $\Delta_{2,\mathrm{max}}(t)$ from Theorem 2 to be the corresponding errors for $g_i=\nabla f_i(w^t),\forall i\in[m]$ for any $t>0$.

**Assumption 5** (Bounded domain). *The set $\mathcal{W}$ is closed and convex with diameter $R^2$.*

---

**Algorithm 6** Distributed Projected Gradient Descent with SparseReg compressor

---

**Require:** Initial iterate $w^0 \in \mathcal{W}$, Step size $\gamma > 0$

  Server
  SparseReg-`Init()`
  **for** $t = 0$ to $T-1$ **do**
    Send $w^t$ to all clients $i \in [m]$.
    Receive $\tilde{b}_i^t$ from clients $i \in [m]$.
    $\tilde{g}^t \leftarrow$ SparseReg-`Decode`($\{\tilde{b}_i^t\}_{i \in [m]}$)
    $w^{t+1} \leftarrow \text{proj}_{\mathcal{W}}(w^t - \eta_t \tilde{g}^t)$
  **end for**
  Client(i) at iteration $t$
  Receive $w^t$ from server.
  $\tilde{b}_i \leftarrow$ SparseReg-`Encode`($\nabla f_i(w^t)$)
  Send $\tilde{b}_i^t$ to server.

---

**Assumption 6** (Convexity). *The objective function $f(w)$ is convex $\forall w \in \mathcal{W}$.*

We now state our convergence result.

**Theorem 6.** *Under Assumptions 4, 5, 6, running Algorithm 6 for $T$ iterations with step size $\eta_t = \frac{R}{B\sqrt{T}}$, with probability $1 - 2m^2 LT \exp(-d\delta_1^2/8) - mT\left(\frac{L^{2\delta_2}}{\log L}\right)^{-m}$ we have,*

$$\mathbb{E}[f(\bar{w}^T)] - f(w^\star) \leq \frac{R(2B^2 + \Gamma_1)}{2B\sqrt{T}} + \sqrt{\Gamma_1}R, \quad where, \quad \bar{w}^T = \frac{1}{T}\sum_{t=0}^{T-1} w^t$$

$$\Gamma_1 = B^2 \left(1 + \frac{10\log L}{d}\exp\left(\frac{m\log L}{d}\right)(\delta_1 + \delta_2)\right)^2 \left(1 - \frac{2\log L}{d}\right)^m, \tag{11}$$

$$\Gamma_2 = \max_{t \in \{0,1,\dots,T-1\}} \min\{\Delta_{\text{reg}}(t), \Delta_{2,\max}(t)\}$$

From the above theorem, we can see that the high probability terms and $\Gamma_1$ and $\Gamma_2$ are obtained from Theorem 2. Note that $\Gamma = \mathcal{O}(B^2 \exp(-m/d))$, therefore, for large $m$, the additional bias term of $R\sqrt{\Gamma_1}$ is very small. Further, the term $\Gamma_2 \leq B^2$, therefore, $\Gamma_2$ only affects constant terms in the convergence rate due to $\sqrt{T}$ in the denominator. If $\exp(-m/d) = \mathcal{O}(1/\sqrt{T})$ or $m = \Omega(d\log T)$, the final convergence rate of Algorithm 6 is $\mathcal{O}(RB/\sqrt{T})$ which is the rate for distributed GD without compression.

We provide the proof for the above theorem, which modifies the proof of (Bubeck, 2015, Theorem 3.2) to handle a biased gradient oracle. We can also extend our analysis to other function classes, for instance strongly convex functions, by using existing works on biased gradient oracles Ajalloeian & Stich (2020). Extending the proof to FedAvg from distributed GD would require using biased gradient oracles in Li et al. (2020). Further, these proofs can also be extended to HadamardMultiDim compressor, with an additional $\sqrt{d}$ factor in the corresponding error terms from Theorem 1 to account for conversion from $\ell_\infty$ to $\ell_2$ norm.

### G.1 PROOF OF THEOREM 6

At any iteration $t > 0$, we use $\tilde{g}^t$ to denote the estimate of $\nabla f(w^t)$. From the proof of Theorem 2, $\|\mathbb{E}_t[\tilde{g}^t] - \nabla f(w^t)\|_2 \leq \sqrt{\Gamma_1}$, and $\mathbb{V}ar_t(\tilde{g}^t|w^t) \leq \Gamma_2, \forall t > 0$, where $\mathbb{E}_t$ and $\mathbb{V}ar_t$ are the expectation and variance wrt the randomness in the SparseReg compressor at iteration $t$. We take a union bound over the high probability terms in Theorem 2 over all iterations $t = 0$ to $T-1$.

We can write the following equation by convexity of $f(w^t)$.

$$f(w^t) - f(w^\star) \leq \langle \nabla f(w^t), w^t - w^\star \rangle = \langle \tilde{g}^t, w^t - w^\star \rangle + \langle \nabla f(w^t) - \tilde{g}^t, w^t - w^\star \rangle$$

$$\leq \frac{1}{2\eta}(\|w^t - w^\star\|_2^2 - \|w^t - \eta\tilde{g}^t - w^\star\|_2^2) + \eta\|\tilde{g}^t\|_2^2/2 + \langle \nabla f(w^t) - \tilde{g}^t, w^t - w^\star \rangle$$

In the second line, we use $2\langle a, b \rangle = ||a||_2^2 + ||b||_2^2 - ||a-b||_2^2$. Now, taking expectation wrt the randomness in SparseReg at iteration $t$, we obtain,

$$\mathbb{E}_t[f(w^t)] - f(w^\star) \leq \frac{1}{2\eta}(||w^t - w^\star||_2^2 - \mathbb{E}_t[||w^t - \eta\tilde{g}^t - w^\star||_2^2]) + \eta\mathbb{E}_t[||\tilde{g}^t||_2^2]/2$$

$$+ \langle \nabla f(w^t) - \mathbb{E}_t[\tilde{g}^t], w^t - w^\star \rangle$$

$$\leq \frac{1}{2\eta}(||w^t - w^\star||_2^2 - \mathbb{E}_t[||w^{t+1} - w^\star||_2^2]) + \eta(||\mathbb{E}_t[\tilde{g}^t]||_2^2 + \mathbb{V}ar_t(\tilde{g}^t))/2$$

$$+ ||\nabla f(w^t) - \mathbb{E}_t[\tilde{g}^t]||_2 \cdot ||w^t - w^\star||_2$$

$$\leq \frac{1}{2\eta}(||w^t - w^\star||_2^2 - \mathbb{E}_t[||w^{t+1} - w^\star||_2^2]) + \eta(B^2 + \Gamma_2)/2 + \sqrt{\Gamma_1}R$$

In the second line, we use the non-expansiveness of projections on a convex set, $||w^t - \eta\tilde{g}^t - w^\star||_2 \geq ||\text{proj}_{\mathcal{W}}(w^t - \eta\tilde{g}^t - w^\star)||_2$, the decomposition of $2^{nd}$ moment into square of mean and variance, and cauchy-schwartz inequality. In the third line, we plug in bounds of $\Gamma_1, \Gamma_2$, diameter of the set and by triangle inequality, argue that $\mathbb{E}[\tilde{g}^t]$ also lies in an $\ell_2$ ball of radius $B$.

Finally, we take expectations wrt all random variables, unroll the recursion from $t=0$ to $T$, and divide both sides by $T$.

$$\frac{1}{T}\sum_{t=0}^{T}\mathbb{E}[f(w^t)] - f(w^\star) \leq \frac{R^2}{2\eta T} + \frac{\eta(B^2 + \Gamma_2)}{2} + \sqrt{\Gamma_1}R \leq \frac{R(2B^2 + \Gamma_1)}{2B\sqrt{T}} + \sqrt{\Gamma_1}R$$

We obtain the final inequality by plugging in the step size $\eta = \frac{R}{B\sqrt{T}}$. By convexity of $f$, for $\bar{w}^T = \sum_{t=0}^{T-1}w^t$, we obtain,

$$\mathbb{E}[f(\bar{w}^T)] - f(w^\star) \leq \frac{1}{T}\sum_{t=0}^{T-1}\mathbb{E}[f(w^t)] - f(w^\star) \leq \frac{R(2B^2 + \Gamma_1)}{2B\sqrt{T}} + \sqrt{\Gamma_1}R$$

