# OpenReview forum: "Collaborative Compressors in Distributed Mean Estimation with Limited Communication Budge"
_ICLR.cc/2025/Conference — Submitted to ICLR 2025_

### Official Review · Reviewer_ZsMk · 2024-10-28

**Soundness:** 3
**Presentation:** 3
**Contribution:** 3
**Rating:** 6
**Confidence:** 3

**Summary:**

The paper addresses the problem of high-dimensional mean estimation. It introduces several collaborative compression schemes designed to exploit similarities among vectors in distributed settings. These schemes come with error guarantees across various metrics, including $\ell_2$-error, $\ell_{\infty}$-error, and cosine distance, with error rates improving as the number of clients increases and degrading with greater dissimilarity among clients. In low dissimilarity scenarios, the proposed methods can outperform several baseline approaches.

**Strengths:**

- The work addresses a relevant problem in distributed machine learning. It provides error guarantees for different error metrics, which adds robustness to its evaluation.

- The proposed methods can benefit from client similarity, offering a potential advantage for applications with close-to-homogeneous data distributions. This is reflected both in theory and in experiments.

**Weaknesses:**

- The proposed schemes are designed to work in the setting where the data is close to homogeneous, and perform worse when clients have high dissimilarity. In such settings, the introduced methods do not perform as well as baseline approaches. This significantly limits their usefulness in many real-world application, e.g., federated learning, where client data is often highly heterogeneous.

**Questions:**

- It is unclear in Section 3 what parts are original contributions by the authors versus prior work. The abstract mentions three new schemes, while the conclusion refers to four, which makes the contributions hard to identify. Could the authors clarify this point?
- Is "Technique I" from Section 3 actually introduced in the paper?

---

> ### Author Response · Authors · 2024-11-21
> **Response**
>
> We thank the reviewer for their review. We address all the weaknesses and questions.
>
> **Weaknesses**
>
> 1. [Heterogeneity] The performance of our schemes degrade with increase in dissimilarity between client vectors $g_i$. However, in Section 2.1, we provide an example where the maximum dissimilarity between clients, $\max_{i,j\in [m]}\|\|g_i - g_j\|\|$, is large, however, the average dissimilarity between clients is small for large $m$ as $\Delta_{\infty} \approx \frac{c}{m}\max_{i,j}\|\|g_i - g_j\|\|$, where $c$ is a constant smaller than $c \ll m$. In such scenarios, our methods also perform well for large $m$, as $\Delta_{\mathrm{Hadamard}} \approx \sqrt{\frac{c}{m}}\max_{i,j\in [m]}\|\|g_i - g_j\|\|$, where $\Delta_{\mathrm{Hadamard}}$ has been defined in Theorem 1.
>
>
> **Questions**
>
> 1. [New results] We have revised the abstract to mention our $4$ proposed schemes --  NoisyGD (Algorithm 1), HadamardMultiDim (Algorithm 3), SparseReg (Algorithm 4) and OneBit (Algorihtm 5). In Section 3, our contribution lies in modeling the problem of distributed mean estimation for the unit vector as halfspace learning with noise, Lemma 1. Techniques I and II are off-the-shelf techniques  for halfspace learning with noise, and corresponding Theorems 1 and 2 apply their results to our problem setting.
> 2. Technique I has not been explicitly introduced in the paper as it is an existing algorithm for solving halfspace learning with noise. In Lines 455-460, we provide it's brief explanation.
>
>
>
> We are happy to answer any additional questions or clarifications that the reviewer may have.

---

> > ### Author Response · Authors · 2024-11-26
> >
> > We thank the reviewer for their review and request that they ask any follow-up questions or clarifications that they may have.

---

### Official Review · Reviewer_5FAd · 2024-10-29

**Soundness:** 3
**Presentation:** 1
**Contribution:** 2
**Rating:** 5
**Confidence:** 4

**Summary:**

The paper considers the distributed mean search problem. This may be applicable, for example, when gradients are averaged in distributed learning. The authors present new collaborative compressors for this problem. The authors provide a comprehensive theoretical analysis for the "variance" estimation and small experimental comparison with some competitors.

**Strengths:**

1) Overall the paper is easy to read.

2) The ideas look new and are a quite good contribution.

3) The authors do a pretty good job of giving intuition and explaining the details of the new operators, as well as the theoretical results (physics of the various $\Delta$ and so on)

4) The literature review is not badly done, comparison with other papers is partly inherent, although it seems incomplete - see 1st weakness.

5) I went through the proofs quickly, the results seem correct, and relatively expected.

**Weaknesses:**

1) Estimates in terms of $\Delta_2$ are more interesting than in terms of $B$ as given by the authors. Let me explain. From the point of view of distributed optimization it is important how the method that uses compressed communications converges.

For example, in the paper (Szlendak et al., 2021)  the estimate on PermK is written in terms of $Delta_2$, the MARINA optimization algorithm has good convergence estimates. If we write these estimates in terms of $B$, is it possible to get such good estimates? The paper under review does not answer this question. I will ask an even more general question: how does a distributed gradient descent (or a more advanced method) with compression operators presented in the paper converge?

2) The experiments are weak.
а) They look quite simple and seem even simpler than in the original paper (Suresh, 2016).

b) Different competitors are used in different experiments. This looks strange and suspicious.

с) Tuning algorithms are not described (maybe I didn't look carefully).

d) Obviously, gradient descent is not the most advanced algorithm. The same PermK paper uses MARINA, which is a more advanced algorithm designed for distributed setting with compression.


3) The paper is written in a rush. I'm not the most diligent typos finder, but

a) style of citation - without brackets looks weird

b) "$\ell_2$-error" or $\ell_2 error$?

c) line 333: a dot at the beginning

d) line 523: space 2i

e) Plots design: 1) sometimes there is a dot in the legend (e-i), sometimes there is not (a-d), 2) lines do not start from one point (f,e), 3) axis on (g) is cut off, 4) where is green on (e,h)? 5) legend of (a-c) is smaller than of others

f) may be make sense to put Algorithm 1 into Appendix

g) OpenReview title: Budge

===========================

While I enjoyed the article, the current condition of the paper does not allow for its acceptance.

**Questions:**

1) How do the operators proposed in the paper relate to the uncertainty principle from (Safaryan et al., 2021)? Does the answer to this question depend on the similarity of $g_i$? It seems that it does. How? If all $g_i$ are the same?

2) How do the gradient similarities introduced in the paper relate to the similarities used for example in the paper?

It uses hessian similarities:
Hendrikx, Hadrien, et al. "Statistically preconditioned accelerated gradient method for distributed optimization." International conference on machine learning. PMLR, 2020.

Why I ask, the similarity of the gradients often varies from point to point and may in general be not bounded at all (e.g. solve on $R^d$ two quadratic problems with different matrices = linear regressions with a quadratic loss function on two different datasets). Hessian similarity is better.

---

> ### Author Response · Authors · 2024-11-21
> **Response  - 1/2**
>
> We thank the reviewer for their review. We address all the weaknesses and questions.
>
>
>
> **Weaknesses**
>
> 1. [Estimates in terms of $\Delta$] We agree with the reviewer that estimates in terms of $\Delta$,  are more interesting than those in terms of $B$, as $\Delta$ can be much smaller than $B$. We explain this issue in our Related Works section to establish the superiority of collaborative compressors over independent ones. For our algorithms, however, the coefficient of $B$ is $\exp(-m)$, see Theorems 1 and 2. Therefore, for large  $m$, which is often the case in distributed mean estimation, the contribution of $B$ will be close to negligible and the only error term remaining will be a deviation term, $\Delta_{\mathrm{Hadamard}}$ and $\Delta_{\mathrm{reg}}$ in Theorems 1 and 2 respectively. While these terms are not exactly equal to $\Delta_2$, in section 2.1, we describe problem settings, where these quantities are comparable to $\sqrt{m}\Delta_2$. In these settings, the results of distributed optimization with PermK algorithm can be extended to our algorithms, albeit with an additional $\sqrt{m}$ scaling.
>
>     Further, we have derived convergence rates for distributed GD with SparseReg compressor. See the global response for more details on this.
>
>
>
>
>
> 2. [Experiment] (a) (Suresh et al 2016) only test Power Iteration and KMeans on MNIST dataset and CIFAR10's features. In Figure 2, we provide experiments for Power Iteration and KMeans on MNIST and FEMNIST, a real federated dataset with real heterogeneity. Further, we also provide experiments for estimating error in different norms and for Linear regression. (b) In Appendix F, Lines 1105-1116, we explain why different baselines are used for different plots. To summarize, for different norms (Figure 2(a)-(c)) and different tasks(Figures 2(d) - (i)), we use baselines appropriate for the norm and the task respectively. There are $12$ algorithms in total, $8$ baselines and $4$ proposed methods, and we did not plot all of them in the same plot for clarity. (c) In Appendix F, we provide additional details on experiments. The communication budget for all baselines is tuned to be the same. To tune the hyperparameters for each task, we first run the task without compression for $20$ different random combinations of hyperparameters. We pick the best hyperparameters from this search and use it for all compressors. (d) Our goal is to propose compressors for distributed mean estimation. Federated learning with compression is one application of DME but not the only one. To test if our compressors for DME work in this setting, we used the simplest example of distributed gradient descent. While it is possible to use our compressors with more advanced techniques in iterative distributed optimization, for instance acceleration(Yuan \& Ma, 2020), variance reduction (Condat et al 2022), intermittent use of a compressor(Gorbunov et al 2021) and communicating the model updates instead of full model weights (Haddadpour et al 2019), we would like to emphasize that these techniques are orthogonal to the design of a compressor. Further, investigating the settings where collaborative compressors work with these techniques is an interesting direction for future work.
>
>     Additionally, we are running experiments on more problem settings. See the global response for details on this.
>
> 3.  Typos : We thank the reviewer for pointing out the  typos and we have fixed them in the revised version. Note that the citation style is according to ICLR style files, so we cannot change it. We address the issues related to plots specifically (e) 3) In Figure 2 (d) - (i), we plot the progress from the $1^{st}$ iteration, hence if there is sufficient progress in $1^{st}$ iteration (Fig 2 (f), (i)), the plots do not start from same point. 4) The green in plots Fig 2 (e), (h) correspond to RandKSpatial which has exactly same performance as RandKSpatialProj in this setting and is thus hidden by it.

---

> > ### Author Response · Authors · 2024-11-21
> > **Response  - 2/2**
> >
> > **Questions**
> > 1. [Uncertainty Priciple]
> >     In the language of (Safaryan et al 2021), the ``uncertainty prinnciple''
> >     applies to the error and number of bits of an independent compressor, which obeys either Definition 2.1 (for biased compressors) or Definition 2.3 (for unbiased compressors) in (Safaryan et al 2021). Our collaborative compressors are not independent and don't satisfy either Definition 2.1 or 2.3. Therefore, the uncertainty principle does not apply to our compressors and we can obtain much smaller error than (Safaryan et al 2021). From Table 2, (Safaryan et al 2021), obtains error that scales with $\tilde{B}^2 = \frac{1}{m}\sum_{i=1}^n || g_i ||^2 $, while our collaborative compressors, in particular, SparseReg obtains error that scales as $B\exp(-m) + \Delta_{\mathrm{reg}}$. $\Delta_{\mathrm{reg}}$ is a deviation term between compressed values of $g_i$. So, if  $\|\|g_i\|\|$ is large but the deviation between $g_i$ is smaller than its norm, the error of SparseReg is much smaller than (Safaryan et al 2021). As (Safaryan et al 2021) uses an independent compressor, their error is larger than that of every collaborative compressor, even PermK and Correlated SRQ.
> >
> >     More gennerally, the ``uncertainty principle'' in (Safaryan et al 2021) is  a trivial corollary of rate-distortion theory. For collaborative compression, information theoretic lower bounds are possible - but they will be more complicated, due to the fact that achievable rate regimes of multi-source coding are not well-characterized. Such lower bounds will be an interesting future work.
> >
> > 2. [Comparison with Hendrikx, Hadrien, et al] Consider the  example mentioned in Section 2.1, where a constant number $c$ of clients have objective function  $f_1(w) = \frac{1}{2}(w - w_1^\star)^\top A_1 (w - w_1^\star)$ while the rest have objective function $f_2(w) = \frac{1}{2}(w - w_2^\star)^\top A_2 (w - w_2^\star)$. Here, $A_1, A_2$ and $w_1^\star, w_2^\star$ are the Hessians and minimas respectively for the two clients. Assuming, we perform distributed gradient descent, the vectors $g_1 = A_1(w - w_1^\star)$ and $g_2 = A_2(w - w_2^\star)$. If $\|\|g_1 - g_2\|\|$  is small, then error of our collaborative methods, in this case SparseReg, will be small.
> >     $$
> >     \begin{align*}
> >        \| \|g_1 - g_2\|\|  = \|\|A_1 (w - w_1^\star) - A_2(w - w_2^\star)\|\| \leq \|\|A_1 - A_2\|\|\cdot\|\|w - w_1^\star\|\| + \|\|A_2\|\| \cdot\|\|w_1^\star - w_2^\star\|\|
> >     \end{align*}
> >     $$
> >     Let $\|\|w_1^\star - w_2^\star\|\| \leq \delta_w, \|\|A_1 - A_2\|\| \leq \delta_A$ and $\max\{\lambda_{\max}(A_1),\lambda_{\max}(A_2)\} \leq L$ and all iterates lie in a set of diameter $D$. Then $\|\|g_1 - g_2\|\| \leq \delta_A D + L \delta_w $. Here, $\delta_w$ is the dissimilarity in the minima, $\delta_A$ is the Hessian dissimilarity, and $L$ is the local smoothness constant. From Section 2.1, in this case $\Delta_{\mathrm{reg}}\approx \sqrt{\frac{c}{m}}\|\|g_1 - g_2\|\| \leq \sqrt{\frac{c}{m}}(\delta_A D + L \delta_w)$. Therefore, as long as $\delta_A$ and $\delta_w$ are small, $\Delta_{reg}$ is small and our methods perform well. (Hendrikx et al 2020) corresponds to the special case of $\delta_A \neq 0$, but $\delta_w =0$.
> >
> >
> >
> >
> > We are happy to answer any additional questions or clarifications that the reviewer may have.
> > If our rebuttal has changed the reviewer's assessment of our paper, we request them to update their score to reflect this.
> >
> > **References**
> >
> > - (Suresh et al 2016) Distributed Mean Estimation with Limited Communication. Arxiv.
> > - (Yuan \& Ma, 2020) Federated Accelerated Stochastic Gradient Descent. NeurIPS.
> > - (Condat et al 2022) Provably Doubly Accelerated Federated Learning: The First Theoretically Successful Combination of Local Training and Communication Compression. Arxiv.
> > - (Gorbunov et al 2021) MARINA: Faster Non-Convex Distributed Learning with Compression. ICML.
> > - (Haddadpour et al 2021) Federated Learning with Compression:
> > Unified Analysis and Sharp Guarantees. AISTATS.
> > - (Safaryan et al 2021) Uncertainty principle for communication compression in distributed and
> > federated learning and the search for an optimal compressor. IAA.
> > - (Hendrikx et al 2020) Statistically Preconditioned Accelerated Gradient Method
> > for Distributed Optimization. ICML.

---

> > > ### Comment · Reviewer_5FAd · 2024-11-25
> > >
> > > Thanks to the authors for the response!
> > >
> > > Of my weaknesses, only the one related to typos has been fixed completely. The rest remained essential.
> > >
> > > 1) The $B$ and $\Delta$ differences are still inherent, although the authors discussed this a bit in the response. I think the authors should think about how to rewrite their results in terms of $\Delta$. Also I think that the results in Appendix H can be improved: now the theory does not guarantee convergence at all (there is an additive but as the authors claim often a small factor), this is strange for most of the known operators from the literature. I recommend the authors to take their time and consider the smooth case (not just the bounded gradient) and get a more optimistic theory.
> > >
> > > 2) The general response says we are waiting for experiments.
> > >
> > > 3) Good! Thanks!
> > >
> > > 4) Good! The only thing "a trivial corollary" is not super accurate and polite.
> > >
> > > 5) I don't agree. Hendrikx doesn't say anything about $\delta_w$. That's the thing, he only cares on $\delta_A$, and $\delta_w$ can be anything. And the authors care about the behavior of $\delta_w$, which looks like a limitation compared to existing literature.
> > >
> > > I think the paper still needs to be finalized. I really ask the authors to take their time and do things carefully, they were in a hurry, both with the paper itself and with the response. For me, waiting until the next suitable conference is a good option.

---

> > > > ### Author Response · Authors · 2024-11-26
> > > > **Response to Additional Comments**
> > > >
> > > > 1.  For our compressors SparseReg and HadamardMultiDim, in Section 2.3 and this rebuttal , the quantities $\Delta_{\rm Hadamard}$ and $\Delta_{\rm reg}$, can be expressed in terms of $\Delta_2$ for certain problem settings. The remaining term of $B^2\exp(-m)$ cannot be decomposed into a term of $\Delta$, but the exponential dependence ensures that this is already very small. The only condition when this is not satisfied is when  $\Delta_2 = \mathcal{O}(B^2\exp(-m))$, satisfied for distributions with very strong concentration. As an example, in a 1-dimensional case, all distributions on a bounded set $[-B,B]$ have $\Delta_2 = \mathcal{O}(B^2 m^{-1})$ and therefore, our method performs well for them.
> > > > As for the results in Appendix H, the reviewer asked the following question : ``how does a distributed gradient descent (or a more advanced method) with compression operators presented in the paper converge``. And our results in Appendix H answer it for convex Lipschitz losses. Finding the best compression technique for smooth distributed convex optimization is NOT the goal of this paper, we seek solutions for distributed mean estimation. Distributed optimization is only one application of DME.
> > > >
> > > > As for the additive term in the result in Appendix H, independent biased compressors in literature satisfying nice properties, see Definitions 2,4 and 5 in (Beznosikov et al 2023), do not incur this term. However, biased collaborative compressors, including (Suresh et al 2022) and our compressors,  do not satisfy these properties.  So, we must rely on the convergence of gradient descent with biased gradients where this additive term cannot be removed. In (Ajalloeian & Stich 2020), biased gradients are obtained from $\zeta \neq 0$ in Assumption 4, and from Theorems 4 and 6, we see that the final error has an additive term dependent on this bias.
> > > >
> > > > 2.  We have included experiments in Appendix F.1 on logistic regression. We provide additional details in the comments to our global response.
> > > >
> > > > 5. Note that (Hendrikx et al 2020) do not require $\delta_w = 0$, however, for arbitrarily large value of $\delta_w$, the solution $w^\star = \arg\min_{w} f(w) \triangleq \frac{1}{m}\sum_{i=1}^m f_i(w)$ is bad for every machine, as $f(w^\star)$ can be very large. Consider the example proposed in the rebuttal of two quadratics, with $m-c$ clients having $f_1$ as their objective and $c$ clients having $f_2$ as their objective. Then, the minima $w^\star = A^{-1}(\frac{m-c}{m} A_1 w_1^\star + \frac{c}{m} A_2 w_2^\star)$, where $A = \frac{m-c}{m}A_1 + \frac{c}{m}A_2$. Then, we can write the loss $f(w^\star)$ as
> > > >
> > > > $$
> > > > \begin{align*}
> > > > f(w^\star) = \frac{c(m-c)}{2m^2} (w_1^\star - w_2^\star)^\top A^{-1} (w_1^\star - w_2^\star) \geq \frac{c}{2mL}\delta_w^2
> > > > \end{align*}
> > > > $$
> > > > Therefore, if $\delta_w$ is arbitrarily large, then the loss at the optima $f(w^\star)$ is arbitrarily large. This is undesirable in federated learning as then the optima $w^\star$ is poor for all clients. Even though (Hendrikx et al 2020) don't explicitly state this as an assumption, as they only show convergence of their solution to $w^\star$, a small $\delta_w$ is required for federation. Note that this dependence on $\delta_w$ is similar to that required for our methods to work, i.e., $\frac{\delta_w}{\sqrt{m}}$ must be small. In our rebuttal, we set $\delta_w=0$ so as to compare the heterogeneity condition of (Hendrikx et al 2020) directly with our setup, without worrying about the quality of the solution, $f(w^\star)$.
> > > >
> > > > Please let us know if there are any additional concerns that we may answer.
> > > >
> > > > **References**
> > > > - (Beznosikov et al 2023) On Biased Compression for Distributed Learning. JMLR.
> > > > - (Suresh et al 2022) Correlated Quantization for Distributed Mean Estimation and Optimization. ICML.
> > > > - (Ajalloeian & Stich 2020) On the Convergence of SGD with Biased Gradients. Arxiv.

---

### Official Review · Reviewer_bH5L · 2024-11-03

**Soundness:** 2
**Presentation:** 2
**Contribution:** 1
**Rating:** 5
**Confidence:** 4

**Summary:**

This paper proposes several different collaborative compression schemes. For each schemes this paper proposes corresponding collaborative compression algorithms and theoretically analyse the estimation error on different metrics. This paper conducts experiments on standard dataset to test the algorithms.

**Strengths:**

This paper studies a relatively new direction on distributed optimzation and compression techniques: collaborative compression. The author proposes three different schemes, using different norm to measure the estimate error and provides theoretical analysis.

**Weaknesses:**

1. The literature research is not enough. The author provides comparison of existing independent compression and collaborative compression methods in Table 2. But in the list the independent compression techniques are not the latest achievements. At least, considering error feedback techniques, the accumulated error of independent compression techniques can be bounded and the convergence rate is faster than those proposed in Table 2.
2. The theoretical analysis is too simple. Considering collaborative compression methods are not first proposed by author, the theoretical analysis should be more deep-going. In fact the author only computes the estimate error of the compression algorithms under different  norm. It is lack of novelty.
3. The experiment results are also not enough. I think at least conducting experiments on basic neural networks and standard dataset is necessary, for example, tiny transformers on OpenWebtext. In other words, it is necessary to test the algorithm in practical tasks rather than only in the DME. In practical distributed training, the global gradient is the mean estimate of local ones. Can such algorithms work? I think copying the experimental setting in [1] is not enough.

[1]  Shuli Jiang,Pranay Sharma,and Gauri Joshi.Correlation aware sparsified mean estimation using random projection.

**Questions:**

1. I think at least the basic application of such estimator is necessary to study. For example, the convergence rates (or gradient complexities) of algorithms applied to smooth optimization are necessary in a theory work. Only bounding the estimate error can not illustrate the effectiveness of compression algorithms when applying to specific tasks. Moreover, the communication complexity is also necessary.
2. This work follows [1] but has few novelty. I think this paper needs more deep-going study. The existing content is not sufficient to support publication.

[1]  Shuli Jiang,Pranay Sharma,and Gauri Joshi.Correlation aware sparsified mean estimation using random projection.

---

> ### Author Response · Authors · 2024-11-21
> **Response - 1/2**
>
> We thank the reviewer for their review. We address the weaknesses and questions below.
>
> **Weaknesses**
>
> 1.  [Literature Review] We would like to emphasize that we are proposing new compressors for distributed mean estimation (DME), NOT optimization algorithms for supervised federated learning with compression. Therefore, appropriate baselines for our paper are existing compressors used in DME, which have been covered in Table 2. Further, Table 2 provides the best possible $\ell_2$ estimation errors in literature, NOT convergence rates of federated optimization algorithms with these compressors.
>
>     Iterative federated optimization is one application of our compressors for $\ell_2$ metrics. The compressors we propose, especially for $\ell_\infty$ norm and cosine distance, can be used for applications for which no equivalent to error feedback exists, for instance, distributed KMeans, distributed PCA etc.
>
>     Note that error feedback is a technique used in iterative federated optimization which performs DME in every round. However, this technique and several previously used techniques for iterative distributed optimization, for instance acceleration (Yuan \& Ma, 2020), variance reduction (Condat et al 2022), intermittent use of a compressor (Gorbunov et al 2021) and communicating the model updates instead of full model weights (Haddadpour et al 2019) are orthogonal to the design of a compressor. Algorithmically, these techniques can be performed with most compressors, including ours, however, their theoretical benefits have been established only for simple independent compressors and certain collaborative compressors like PermK (Condat et al 2022).
>
>
>
>     In the revised version, we mention this in our conclusion and leave the extension of error feedback to collaborative compressors as a direction for future work.
>
>
> 2. [Metric for evaluation] For compressors in DME, the correct metric for comparison is the estimation error from the true mean and the number of bits sent, both of which we report in Table 1. Further, we do not compute the estimation error of our algorithms under each norm, rather for each norm ($\ell_\infty, \ell_2$, cosine distance), we propose a collaborative compression algorithm that achieves small estimation error.
>
>     The papers on DME (everything in Table 2, apart from PermK) analyze only the estimation errors of the compressor. Federated learning is just one application of these compressors.
>
>
>
>     The key novelty in all of our compressors is the flexibility of collaboration -- a) In Section 2, full collaboration between clients for encoding obtains $\exp(-m)$ dependence on diameter of space $B$, which is the best in literature, b) In Section 3, full collaboration between clients for decoding recovers unit vector of mean at extremely low communication cost.
>
> 3. [Experiments]
>         We are running additional experiments as suggested by the reviewer. See the global response for more details.
>
>     Note that in  experiments, see Figure 2, we do check our methods on practical tasks (Distributed KMeans, PCA and Linear regression). Further, in Lines 048-053, we describe how any distributed mean estimation algorithm, including the ones we propose, can be used for distributed gradient training. Additionally, we perform distributed gradient descent with compression for linear regression experiments.

---

> > ### Author Response · Authors · 2024-11-21
> > **Response - 2/2**
> >
> > **Questions**
> >
> >
> > 1. [Convergence rates] It depends on what theory the reviewer is talking about.
> >     We point the reviewer to our rebuttal to Weakness 2, for the questions on the appropriateness of estimation error for compressors. These expressions of estimation errors can be plugged in in any analysis of SGD with biased gradients (Ajalloeian \& Stich 2020). Further, we have derived convergence rates for distributed GD with SparseReg compressor. See the global response for more details on this.
> >
> >     As for communication complexity, we have provided the number of bits required for each our compressors and baselines in Tables 1 and 2.
> >
> >
> >
> >
> >
> > 2. [Novelty beyond Jiang et al.] Note that for RandKSpatialProj, proposed in (Jiang et al 2023), the theoretical guarantees hold only if the correlation between the vectors is already known. This is a strong assumption, and in practice (Jiang et al 2023) use a heuristic value for correlation. In contrast, our methods do not require any additional knowledge about the vectors for our theoretical guarantees to hold. Further,  (Jiang et al 2023) can only guarantee $\ell_2$ error, while we provide algorithms for $\ell_2, \ell_\infty$ and cosine distance. In addition to this, our experiment setup differs from (Jiang et al 2023) in the dataset we use for Distributed Power Iterations and Distributed KMeans. We use the real federated dataset FEMNIST, while (Jiang et al 2023) split the simpler Fashion-MNIST dataset into clients. In all our experiments as well, Figure 2, our methods outperform (Jiang et al 2023).
> >
> >
> > We are happy to answer any additional questions or clarifications that the reviewer may have.
> > If our rebuttal has changed the reviewer's assessment of our paper, we request them to update their score to reflect this.
> >
> >
> > **References**
> >
> > - (Ajalloeian \& Stich 2020) On Convergence of SGD with Biased Gradients. Arxiv.
> > - (Yuan \& Ma, 2020) Federated Accelerated Stochastic Gradient Descent. NeurIPS.
> > - (Condat et al 2022) Provably Doubly Accelerated Federated Learning: The First Theoretically Successful Combination of Local Training and Communication Compression. Arxiv.
> > - (Gorbunov et al 2021) MARINA: Faster Non-Convex Distributed Learning with Compression. ICML.
> > - (Haddadpour et al 2021) Federated Learning with Compression:
> > Unified Analysis and Sharp Guarantees. AISTATS.
> > - (Jiang et al 2023) Correlation aware sparsified mean estimation using
> > random projection. NeurIPS.

---

> > > ### Author Response · Authors · 2024-11-26
> > >
> > > We thank the reviewer for their review and request that they ask any follow-up questions or clarifications that they may have.

---

> > > > ### Comment · Reviewer_bH5L · 2024-12-03
> > > >
> > > > Thanks to the authors for their patient reply, which has solved my concerns to some extent. I have revised my score. But overall, I still think this paper needs more detailed touches and modifications. Best regards.

---

### Official Review · Reviewer_RnXB · 2024-11-04

**Soundness:** 3
**Presentation:** 2
**Contribution:** 2
**Rating:** 5
**Confidence:** 3

**Summary:**

This paper proposes a sequence of collaborative compressors for distributed mean estimation problem, exploiting the observation that the local vectors can share similarities with each other. The authors design one compression scheme for each similarity metric among $\ell_\infty$ norm, $\ell_2$ norm, and cosine similarity. Their respective upper bounds on estimation error is also provided, decaying with the number of clients $m$ and relying on the difference between local vectors. Experiments on distributed mean estimation, KMeans, power iteration, and linear regression are conducted.

**Strengths:**

* It is an interesting problem to exploit the similarities between local vectors for compression.
* The paper considers multiple similarity metrics and provides rigorous analysis on the estimation error bound for each of the compression scheme.
* The presentation of this paper is mostly clear to me.

**Weaknesses:**

* The error bounds developed in this paper have an extra term $\Delta$ that does not decay with $m$. On the contrary, previous works as summarized in Table 2 of this paper can avoid this problem. This theoretical gap is confusing to me and I look forward to authors' explanations.
* The experiments on gradient aggregation are performed only for linear regression tasks. I would suggest working on logistic regression or neural networks to better demonstrate the performance of the proposed schemes in modern machine learning settings.

**Questions:**

* There is a typo at your openreview submission title.

* "However, independent compressors suffer from a significant drawback, especially when the vectors to be aggregated are similar/not-too-far, which is often the case for gradient aggregation in distributed learning." I'm not sure if this is the case, because data heterogeneity (which leads to heterogeneous local gradients) seems to be a major challenge discussed in literature. I hope the authors can further clarify on this.

* In Figure 2(a): The blue curve (RandK) is missing.

* In Figure 2(i): What does the y-axis stand for? Why do the curves not converge and not descend too much (from 1.14e4 to 1.06e4)? Will you need more iterations for this experiment?

* In caption of Table 1: I suggest the authors do not cite Wikipedia articles in academic writing.

* In Line 265: Is it $j\in[m]$ or $j\in[d]$?

* In Line 921: There is one more $+$ sign in the equation.

---

> ### Author Response · Authors · 2024-11-21
> **Response**
>
> We thank the reviewer for their review. We address all the weaknesses and questions.
>
> **Weaknesses**
>
> 1.  [Deviation Term not going to 0]
>     This is actually a trade-off of using the correlation between vectors for compression. When the deviation term $\Delta$ is close to 0, collaborative compression is very effective since the other factor can decay as $\exp(-m)$. However in the schemes of Table 2, the decay is at best $O(1/m).$ In addition, independent compressors in Table 2 have dependence on $B$, the diameter of the space, and $B\gg \Delta$. Among the collaborative compressors in Table 2, RandKSpatial and RandKSpatialProj require the exact correlation between all vectors for their guarantees to hold. As for the remaining two correlated compressors in Table 2, PermK and CorrelatedSRQ, our proposed algorithms outperform them under the certain conditions, which are discussed in the paper:
>     In Section 2.5, through a real example, we establish these conditions for our algorithm HadamardMultiDim, with a similar extension to SparseReg. Further, in Appendix B, Lines 792-802, we establish conditions under which any $\ell_2$ compressor, which includes all baselines in Table 2, performs worse than OneBit for unit vector estimation.
>
>
>
>
> 2. [Experiments]
>     We are running some experiments in these directions. See the global response for more details. Since our contribution is in the mean estimation problem, and not the subsequent downstream task of optimization, we skipped such experiments so as to keep the focus on contribution.
>
>
> **Questions**
>
>
> 1. Thanks for pointing this out. We will fix this in the title.
> 2.  Yes, data heterogeneity is a major challenge in federated settings. However, such heterogeneity is not unbounded, otherwise there will not be any utility of doing federated learning.
>
>     In particular, there are still settings, where the maximum data heterogeneity between two clients is large, but it is low on average.
>     In such settings, using collaborative compressors, in particular, our methods, is advantageous over independent compressors. One example is mentioned in Section 2.3, where a small number of clients, $c \ll m$, have a different distribution than the rest of the clients. These two distributions, corresponding to the two sets of clients, can be very far from each other. However, the average pairwise distance between clients' distributions is small, and thus our methods achieve low error.
>
>
> 3.  For isotropic gaussian vectors in Figure 2 (a), RandKSpatial has no advantage over RandK. Therefore, their plots overlap in this case.
> 4.  The y-axis stands for Test MSE similar to Figure 2(f). As the responses $y$ for the synthetic dataset are generated with gaussian noise, the minimum test MSE is not $0$. Therefore, the plots are raised by large constants, which makes the decrease appear not large enough. As the reviewer correctly points out, running for more iterations will decrease the error further. If we plot the $\ell_2$ error of the trained model, it should converge to $0$.
> 5. We cited the Wikipedia article for the error function. We have changed it in the revised version.
> 6. We have fixed the typos in Lines 265 and 921 in the revised version.
>
>
> We are happy to answer any additional questions or clarifications that the reviewer may have.
> If our rebuttal has changed the reviewer's assessment of our paper, we request them to update their score to reflect this.

---

> > ### Author Response · Authors · 2024-11-26
> >
> > We thank the reviewer for their review and request that they ask any follow-up questions or clarifications that they may have.

---

> > ### Comment · Reviewer_RnXB · 2024-11-26
> >
> > Thanks the author for the detailed response.
> >
> > “When the deviation term $\Delta$ is close to 0”: Theoretically, in order to outperform the existing methods, we have to assume $\Delta=o(1/m)$. This is the regime where we can say $\Delta$ is small rigorously. The authors may want to explore whether there is a reasonable data generating model that leads to this regime. Or, whether some algorithm can lead to an extra term like $\Delta/m$ rather than $\Delta$.
> >
> > I tend to keep my current score and encourage the authors to improve the current version of this paper.

---

> > > ### Author Response · Authors · 2024-11-26
> > > **Response**
> > >
> > > In Section 2.3, we provide an exact example where $\Delta = o(1/m)$.

---

> > > > ### Comment · Reviewer_RnXB · 2024-11-26
> > > >
> > > > From what I can gather, you seem to provide examples where $\Delta=O(1/\sqrt{m})$. Please clarify if I got it wrong.

---

> ### Author Response · Authors · 2024-11-28
>
> There seems to be some confusion about which $\Delta$ needs to be $o(1/m)$. Note that all $\|\cdot\|$ are $\ell_\infty$ norm. For our example in Section 2.3, $\Delta_\infty = \Theta(\frac{\max_{i,j\in [m]} \|\|g_i - g_j\|\|}{m})$ and the error of method depends on $\Delta_{\rm Hadamard} = \Theta(\frac{\max_{i,j\in [m]}\|\|g_i - g_j\|\|}{\sqrt{m}})$. From the discussion in Section 2.3, specifically lines 411-412, we require $\Delta_{\infty} < \mathcal{O}(dB/(mK))$ for our methods to outperform CorrelatedSRQ. From the expression of $\Delta_{\infty} = \Theta(\frac{\max_{i,j\in [m]} \|\|g_i - g_j\|\|}{m})$ this is possible for small values of $\max_{i,j\in [m]} \|\|g_i - g_j\|\|$.
>
> We hope this clarifies the reviewer's concerns as we do have cases where our methods outperform all baselines even theoretically.

---

### Author Response · Authors · 2024-11-21
**Global response : Changes in the revision**

We thank the reviewers for their insightful feedback. We have uploaded a revised manuscript incorporating the changes suggested by reviewers. These changes are listed below.

1. **Convergence rate of SparseReg for Lipschitz convex functions** (Reviewers bH5L and 5FAd): In Appendix G, Theorem~6 provides convergence rates for Distributed Gradient Descent using our compressor SparseReg.  We obtain rates of the order of $\mathcal{O}(\frac{RB}{\sqrt{T}} + RB\exp(-m/d))$, where the first term corresponds to the rate without compression. As we can see the additional error term is extremely small, and for a mild condition on number of machines, $m = \Omega(d\log T)$, it matches the rates without compression. This proves that our schemes can work for distributed optimization. We have also discussed how to extend these results to more function classes and the FedAvg algorithm.

2. **Typos**(Reviewers RnXb and 5FAd): We have fixed all typos pointed out by the reviewers.

3. **Experiments**:
     Since our contribution is in the mean estimation problem, and not the subsequent downstream task of optimization, we skipped extensive experiments on FL tasks so as to keep the focus on our contribution (also keeping with other distributed mean estimation papers). Nonetheless, we are running  additional experiments on logistic regression and NNs. We will update the manuscript as soon as we get these results.

---

> ### Author Response · Authors · 2024-11-26
> **Experiments in revision**
>
> Dear Reviewers,
>
> We have included experiments on logistic regression on the HAR dataset in Appendix F.1. We report the test accuracy and training loss for the 4 best compressors. 3 of these are our baselines, OneBit, SparseReg and HadamardMultiDim. The closest baseline compressor is RandKSpatialProj, which is the 3rd best in terms of both test accuracy and training loss.

---

### Meta-Review · Area_Chair_YBgt · 2024-12-16

**Metareview:**

Following the rebuttal and discussion, the consensus remains more towards (weak) rejection, with the required modifications being a bit too non-minor to do without further reviewing.  Some issues highlighted include technical novelty, limited understanding of when the guarantees beat existing ones and how strong the assumptions are to make that happens, limited experimental scope, and general writing quality.  I hope that these reviews help the authors to further improve the paper.

**Additional Comments On Reviewer Discussion:**

The three reviewers with score below the acceptance threshold all ended up maintaining that being the case.

---

### Decision · Program_Chairs · 2025-01-22

Reject